# Cell loss disrupts mechanical homeostasis to drive retinal pigment epithelium ageing-like phenotype in vitro

Teodora Piskova [1], Aleksandra N. Kozyrina[2], Giedrė Astrauskaitė[2], Mohamed Elsafi Mabrouk [3], Sebastian Schepl[1], Stacy Lok Sze Yam [1], Ragul Ravithas[1], Wolfgang Wagner [3], Massimo Vassalli [2] & Jacopo Di Russo [1,4] ✉

Tissue homeostasis emerges from mechanical feedback loops balanced by cell loss and proliferation, a balance that in postmitotic tissues must be maintained without compensatory proliferation. Yet how these tissues preserve mechanical homeostasis and how this challenges function in ageing remains unclear. To establish the relationship between cell density, mechanical homeostasis, and function, we induced age-mimicking cell loss in a postmitotic retinal pigment epithelium (RPE) in vitro. This model recapitulates key structural hallmarks of RPE ageing, including reduced cell height, shortened microvilli and cytoskeletal reorganisation. The density-reduced RPE establishes a new mechanical equilibrium characterised by tissue stiffening and increased junctional contractility. Functionally, these monolayers exhibit impaired phagocytosis of photoreceptor outer segments due to compromised apicolateral plasticity, which is mechanistically linked to the modulation of actin nucleators, Arp2/3 and formins. Altogether, our findings show that a cell loss-induced shift in mechanical homeostasis drives age-related RPE dysfunction, demonstrating that structural remodelling and mechanics alone can compromise tissue function in ageing.

The structure and form of a biological entity determine its function, as described by the form-function paradigm[1,2]. This paradigm applies across various length scales, from the molecular[3,4] and sub-cellular levels[5,6] to the tissue and organ levels, where proper architecture and cellular connectivity are essential for functionality[7–9]. In recent decades, alongside form and organisation, increasing recognition as key parameters influencing function and homeostasis has also been given to the mechanical properties of tissues[10–12]. These properties are determined by the cellular organisation and environment[13–15], and their regulation within a preferred range engages mechanosensitive feedback loops that sustain physiological function[16–18], a concept referred to as mechanical homeostasis[19–21]. Despite its importance, mechanical homeostasis remains poorly understood in postmitotic tissues such as the central nervous system and the retina, which must adapt structurally to naturally occurring cell death to maintain the tissue's integrity and functional state[22–25].

An emblematic example of this adaptation is the age-related changes occurring in the retinal pigment epithelium (RPE)[26]. In the retina, the RPE is located between the light-detecting photoreceptor cells and the choroid capillaries and provides vital metabolic support to photoreceptor cells[27]. In ageing, this support is challenged by the steady rate of apoptosis in RPE[28–30], which needs to be actively

[1]Institute of Molecular and Cellular Anatomy, RWTH Aachen University, Aachen, Germany. [2]Centre for the Cellular Microenvironment, James Watt School of Engineering, University of Glasgow, Glasgow, UK. [3]Institute of Stem Cell Biology, University Hospital of RWTH Aachen, Aachen, Germany. [4]DWI-Leibniz-Institute for Interactive Materials, Aachen, Germany. ✉e-mail: jdirusso@ukaachen.de

compensated by the remaining cells, leading to major structural changes of the tissue. These include monolayer thinning, with the average RPE cell height reported to decrease, while cell area increases[29,31,32]. Furthermore, the RPE actin cytoskeleton remodels, leading to local fibres thickening or thinning with the formation of transcellular stress fibres[33]. Finally, ageing has been shown to be associated with changes in RPE apical microvilli, which become shorter and sparser[31,34]. In temporal correlation to these structural changes, the functional ability of the RPE to support photoreceptor cells changes towards a reduced ability to phagocyte and process photoreceptor outer segments (POS) fragments, with important implications for retinal degenerative pathologies[35,36].

While it has been proposed that the age-related changes in RPE organisation stem from a time-associated reduction in cell density, direct evidence supporting this hypothesis remains elusive. Moreover, whether such alterations drive a shift in tissue mechanical homeostasis that implies functional decline remains a critical unresolved question. Notably, we have recently shown that the strength of both cell-substrate and cell–cell adhesion forces in RPE cells directly relates to their ability to phagocytose POS, with higher actomyosin contractility leading to reduced phagocytic activity[21]. Finally, because rodent models and human post-mortem tissue studies cannot reveal the precise effects of reduced cell density on monolayer structure from other ageing-related processes, such as oxidative stress or extracellular matrix (ECM) remodelling, it is necessary to establish a reductionist, human-relevant in vitro model.

To address this knowledge and technical gap, we present an optimised in vitro model that employs large-scale apoptosis on demand to promote a reduction in cell density in postmitotic human induced pluripotent stem cell-derived RPE (iRPE). This density-reduced iRPE (dr-iRPE) displays structural changes such as enlarged and thinner cells with shortened apical microvilli and cytoskeleton reorganisation, alongside increased tissue stiffness, enhanced junctional contractility and a remodelled apical cortex. The mechanical changes coincide with impaired POS phagocytosis, reflected by reduced apicolateral deformation, decreased number of internalised particles and increased fragment size. Pharmacological modulation of actin nucleation allowed us to partially rescue function, suggesting that functional alteration may result from a shift in force balances within the RPE that prioritises lateral structural reinforcement instead of apical plasticity needed for POS phagocytosis.

## Results
### Cell density reduction in vitro mimics ageing RPE structure
Inspired by the natural reduction in cell density in RPE in vivo[28,30], we aimed to replicate cell density reduction in vitro by inducing large-scale apoptosis and to study the adaptation of the epithelium to the cell loss. To model a native RPE monolayer, iRPE was employed as a recognised and reliable model for the native tissue[37–40]. Importantly, to investigate cellular force balance, iRPE were cultured on 4 kPa soft polyacrylamide hydrogels with basement membrane-like coating, thereby recapitulating the physiological mechanical and biochemical cues of the native substrate as previously characterised[21] (Supplementary Fig. 1a). In these conditions, iRPE assumes a characteristic honeycomb-like organisation, apicobasal polarisation (Supplementary Fig. 1b), and arrest of proliferation (0.01 to 0.41% within 24 h) (Fig. 1A and Supplementary Fig. 1c). To facilitate on-demand apoptosis, on the 10th day of culture we transduced the cells with an adeno-associated virus (AAV) carrying a construct designed to promote the expression of FKBP-caspase 8 (Fig. 1b, Supplementary Fig. 1d). FKBP-caspase 8 is pro-apoptotic fusion protein that can be activated on demand through AP20187-induced dimerisation, initiating the cellular suicide programme[41]. Therefore, after reaching a stable expression, on the 15th day, we incubated the monolayer with the compound AP20187, inducing monolayer-scale apoptosis that led to the extrusion of 2% of

the cells within the observation time (Fig. 1b, c, Supplementary Fig. 1e and Supplementary Movie 1). By the 20th day (96 h after apoptosis completion), the iRPE monolayer showed a reduction of cell density of approximately 8% (Fig. 1d), which was not compensated by increased proliferation. In fact, we quantified significantly lower proliferation rates compared to control monolayers (Fig. 1a). As a result, the obtained monolayer was referred to as density-reduced iRPE or dr-iRPE.

Morphologically, dr-iRPE presents cells with a larger area on average (Fig. 1e, Supplementary Fig. 1f and g) without experiencing any fluidification or unjamming, as evident by cellular area distribution and shape factor (Supplementary Fig. 1h and i), recapitulating morphological changes observed in human RPE ageing[29]. Furthermore, we noticed that one-tenth of cells within dr-iRPE presented transcellular stress fibres (Fig. 1f), which were also reported to occur frequently in human RPE ageing[33]. Next, we quantified the cell height in fixed and mounted samples (Fig. 1g, h), showing that dr-iRPE is significantly thinner than the controls (9.1 vs 11.6 μm), in agreement with observations from the ageing RPE[31,42]. Lastly, motivated by the documented changes of microvilli size and number in the ageing RPE in vivo[31,34], we evaluated ezrin as a polarity marker and microvilli organiser[43–45]. Qualitative analyses of ezrin staining revealed a distinct morphology of the cellular apices between dr-iRPE and control (Fig. 1i), which can also be quantitatively captured by the reduced protein level detected in the cells by western blot (Fig. 1j). Moreover, the examination of the microvilli length by confocal microscopy showed approximately one-third shorter microvilli in the dr-iRPE compared to the control (Fig. 1k). Based on the studies performed on Ezrin[-/-] mice[43], these observed differences strongly suggest that in the dr-iRPE conditions, the microvilli are less elaborated and more disorganised. Finally, the quantification of the active form of ezrin (phosphorylated ezrin), which connects the actin cortex to the membrane[46], revealed a significant decrease in the dr-iRPE (Fig. 1l).

To exclude the influence of the viral delivery on the observed structural changes, we included a matched viral control, which promoted the expression of GFP (GFP-iRPE) (Supplementary Fig. 2a). Compared to this control, the dr-iRPE was still showing a reduced cell density and cell height (Supplementary Fig. 2b–d), as well as significant differences in both organisation and length of ezrin-positive microvilli (Supplementary Fig. 2e). Next, we addressed the observation that despite most of the cells within the monolayer expressed the reporter after transduction, only a few of the cells ultimately underwent apoptosis during the AP20187 treatment (Supplementary Movie 1). To exclude any possible effects of the leftover recombinant protein in the remaining cells, we included an additional mixed control in the analyses, consisting of an iRPE monolayer made by a mixed population of transduced (10%) and naïve cells. This condition allowed us to target the caspase activation only on a subpopulation of cells (Supplementary Fig. 2f, g). Interestingly, with this approach, apoptosis still caused a significant cell density and height reduction, confirming that these features are independent of the possible presence of inactive recombinant protein in the cells (Supplementary Fig. 2h, i). Lastly, we asked whether the changes were induced by secondary factors released in the medium after the large-scale apoptosis induction[47]. Hence, we transferred a medium pre-conditioned by the dr-iRPE sample during and after apoptosis onto control monolayers (Supplementary Fig. 2j). Nonetheless, the quantification still showed no effect of the "conditioned medium" on cell morphology, density, or height (Supplementary Fig. 2k–m).

Altogether, we present an in vitro model based on cell density reduction via on-demand apoptosis, and we demonstrate that the RPE monolayer responds to large-scale apoptosis with structural adaptation. This includes the formation of transcellular actin fibres, reduced cell height, larger cell area and shorter and remodelled microvilli. Interestingly, these cellular responses appear to result from intrinsic

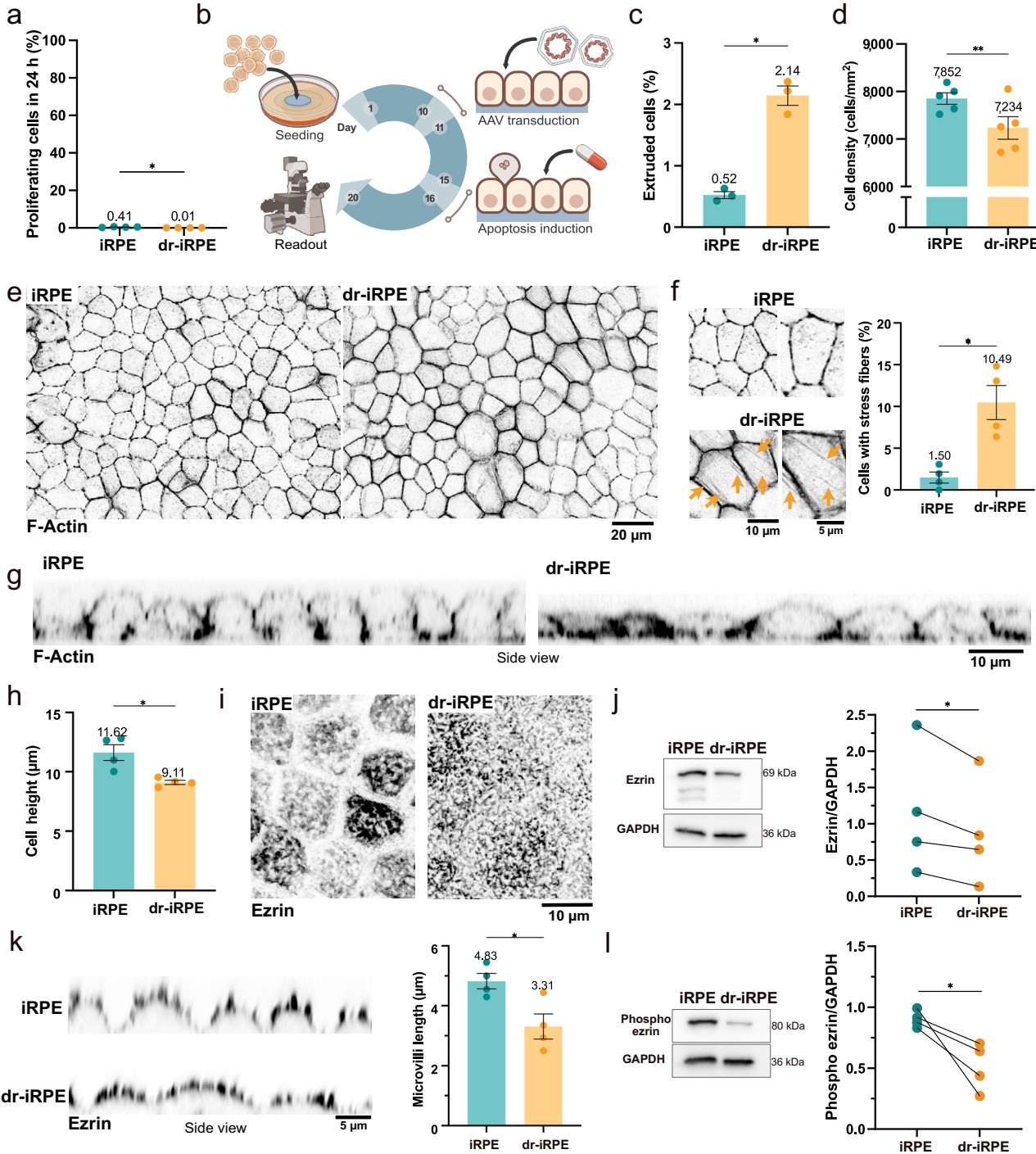

tissue stretching and preservation of cell volume rather than from cellular hypertrophy. Finally, our model recapitulates key morphological features of the aged RPE in vivo, suggesting a causal relationship between cell loss and the structural changes observed in ageing.

**Cell density reduction compromises RPE phagocytotic activity**
One of RPE's main functions in vivo is the daily engulfment and internalisation of oxidised fragments of photoreceptor outer segments (POS), often referred to in the literature as POS phagocytosis[48] or, due to the active nature of nibbling involved, as trogocytosis[49]. Along the lines of the form-function paradigm, we asked whether the structural adaptation observed in response to cell density reduction would alter iRPE's ability to internalise POS. Thus, following an

established assay[21,40], we incubated control and dr-iRPE with fluorescently labelled porcine POS fragments in the presence of bridging molecules Protein-S and MFG-E8 for 4 h (Fig. 2a). During this time, some of the fragments bound to the apical surface of the iRPE and some got internalised by the cells (Fig. 2a, b). After 3D segmentation, we used the apical actin signal to differentiate between bound and internalised fragments (Supplementary Fig. 3a). Quantification revealed that the dr-iRPE exhibited significantly fewer internalised particles, both within the field of view or when normalised by cell number (Fig. 2c, Supplementary Fig. 3b). Next, motivated by a reported trend of larger POS phagosomes in the RPE of aged animals[36], we further quantified the volume of the internalised photoreceptor fragments. Surprisingly, we found the average volume of internalised

**Fig. 1 | Structural adaptation of iRPE to cell density reduction. a** Quantification of proliferating cells within the monolayers on day 20 ($n = 4$, $p = 0.0184$). **b** Experimental timeline highlights key steps of the experiment, including cell seeding on hydrogels (day 1), AAV-mediated transduction with FKBP-casp8-encoding plasmid (day 10 to 11), apoptosis induction by the addition of the dimerising drug AP20187 (day 15 to 16), and day of read-out (day 20). Illustration contains NIAID NIH BioArt elements[119–125]. **c** Quantification of extruded cells during the apoptosis induction ($n = 3$, $p = 0.0172$). **d** Quantification of cell density within the monolayers on day 20 ($n = 5$, $p = 0.0086$). **e** Representative optical sections of iRPE and dr-iRPE stained for F-actin, which reveals cell organisation and sizes. **f** Zoomed-in images from the iRPE and dr-iRPE with arrows indicating the local presence of transcellular stress fibers in the dr-iRPE condition (down). Quantification of cells with transcellular fibres (right, $n = 4$, $p = 0.0110$). **g** Orthogonal cross-sections of F-Actin micrographs. **h** Quantification of cell height ($n = 4$, $p = 0.0203$). **i** Monolayer apical signal of ezrin highlighting microvilli organisation. **j** Immunoblots showing ezrin and GAPDH as a housekeeping protein (left) and quantification of total ezrin protein from immunoblots, normalised to GAPDH (right). Paired values from the same experiment are connected with lines ($n = 4$, $p = 0.0438$). **k** Representative orthogonal views of iRPE monolayers stained for ezrin showing the apical microvilli (left) and the quantification of their average length (right, $n = 4$, $p = 0.0365$). **l** Immunoblots showing phosphorylated ezrin and GAPDH as a housekeeping protein (left) and their quantification (right). Paired values from the same experiment are connected with lines ($n = 4$, $p = 0.0446$). Datapoints in **a**, **c**, **d**, **f**, **h**, **j**, **k** and **l** represent average values per independent biological replicate ($\pm$ SEM in case of bar graphs) with n defined as an independently prepared monolayer, obtained from the same iRPE batch and cultured and treated in parallel. Statistical significance in **a**, **c**, **d**, **f**, **h** and **k** was tested using a two-sided paired t-test, where *: $p < 0.05$, **: $p < 0.01$. Statistical significance in **j** and **l** was tested using a two-sided ratio paired t-test, where *: $p < 0.05$. Representative micrographs in **e** and **i** are representative of four independent experiments with similar results. Source data are provided as a Source Data file. iRPE: induced pluripotent stem cell-derived retinal pigment epithelium; dr-iRPE: density-reduced iRPE; AAV: adeno-associated virus.

fragments to be larger in dr-iRPE monolayers (Fig. 2b, d). Regarding the bound particles, neither their number nor their average volume was found to be significantly different (Supplementary Fig. 3c). Similar differences in internalised particle number and volume were observed in the mixed cell population subjected to cell density reduction, with fewer internalised particles and a larger average particle volume (Supplementary Fig. 3e, g). Finally, the medium transfer from the apoptotic-induced samples showed no significant impact on either the internalised particle count or on the fragment volume (Supplementary Fig. 3f, h).

Altogether, these data demonstrate that structurally remodelled dr-iRPE exhibit deficient internalisation of POS fragments, internalising fewer and larger fragments. By phenocopying the phagocytic deficits observed in ageing RPE, this model suggests that reduced cell density alone can drive functional decline in the remaining cells.

### Density reduction alters actomyosin plasticity and mechanics

POS phagocytosis relies on the active remodelling of the apical actin network, which plays a crucial role in fragment internalisation and phagosome size[21,40,49]. Actin dynamics are essential throughout the process, guiding key steps from initial binding and ensheathment to the final scission of the phagosome[48,50]. Thus, we examined the morphology of the apical actin during internalisation, on a few-cell scale in our 3D segmentation and on a larger monolayer scale. In the side view, we noticed the apical actin of control iRPE monolayers experienced a substantial deformation during the internalisation, such as bulging and enlargement of the protruding cell portion above the cell junctions and clear formation of myosin IIB-positive cups (Fig. 2e–h). In contrast, dr-iRPE did not experience much evident deformation, and phagocytotic cups were less resolved (Fig. 2f, g). Next, we noticed that cells in the dr-iRPE displayed a different apical actin topography compared to the control, having an overall smoother surface, as quantified by area per field of view (Fig. 2h). Finally, on the monolayer scale, we noticed that the dr-iRPE frequently presented large actomyosin-rich apical structures (Fig. 2g, i, Supplementary Fig. 4a, b), which were absent in iRPE samples.

These observations suggest that iRPE cells in the density-reduced condition exhibit altered ability in remodelling their actomyosin cytoskeleton when internalising POS fragments. Consequently, we wondered whether the differences stemmed from distinct patterns of actin-cytoskeleton-related signalling. Hence, we examined the myocardin-related transcription factor A (MRTFA), which is known to regulate actin-associated signalling[51]. We observed a short-term translocation of MRTFA into the nucleus at day 17, shortly after the large-scale cell loss, which went back to levels comparable to the control at day 20, or the day of the read-out (Fig. 3a, b). This demonstrates that the cell density reduction leads to a transient transcriptional adaptation, possibly conveying the observed structural remodelling of dr-iRPE. We then examined the mRNA expression of actin-associated proteins after bulk RNA sequencing (Fig. 3c, Supplementary Fig. 5a). Examining more precisely the expression of these proteins showed an array of significantly differentially expressed actin-related genes in the dr-iRPE. Upregulated genes in dr-iRPE contained genes related to actin branching, polymerisation, depolymerisation, and crosslinking (*ARPC2*, *PFN1*, *FHOD3*, *INF2*, *CFL2*, *ACTN1*, *FSCN1*, *FLNA*). Downregulated genes included ezrin and the formins *DIAPH1*, *DIAPH2*, *DAAM1* and *DAAM2* (Fig. 3d, e). Next, to further assess the parallel between our model and ageing RPE, we compared global gene expression changes in iRPE and dr-iRPE with those reported in ageing RPE from two independent studies[52,53] (Supplementary Fig. 5b–d). Considering the significant differences from our reductionist approach and in vivo human data, as well as the differences between the results of the studies, it is worth noting that our model may mimic further aspects of RPE ageing besides the actomyosin remodelling discussed in this work.

Altogether, these insights point to a distinct signature of actin-associated proteins in dr-iRPE that implies altered actomyosin network organisation, consistent with the observed changes in subapical F-actin topography. Such reorganisation may prevent the formation of the prominent apical deformations or bulges required for efficient phagocytosis, instead promoting the emergence of localised actin-rich structures.

As we previously showed that RPE mechanical equilibrium influences the epithelium phagocytotic activity[21], we sought to determine whether the observed actin network remodelling would alter the mechanical properties of dr-iRPE. To characterise the viscoelastic properties of the monolayer as a multicellular system, we conducted nanoindentation using a spherical tip with a diameter of 20 μm, allowing to probe a contact area of approximately 10 μm in diameter (Fig. 4a, Supplementary Fig. 6a, Supplementary Movie 2). This approach allowed us to measure the Effective Young's modulus of the uppermost 2.5 μm of the monolayer, which approximately corresponds to the apical-most 9% of the cell height (average height of live iRPE is $27.8 \pm 0.6$ μm). The Effective Young's modulus of dr-iRPE was roughly one-third higher (694 Pa vs 513 Pa), pointing to a stiffer tissue (Fig. 4b). Next, using the same nanoindentation setup, we quantified both storage and loss moduli at different probing frequencies to characterise the monolayer viscoelasticity[54]. With this approach, we found a significant increase in the storage modulus of the dr-iRPE while the loss modulus remained unchanged (Fig. 4c, Supplementary Fig. 6b). Similar differences were observed when dr-iRPE was compared to the virus control or when the condition with mixed cell population was analysed without or with cell extrusion (Supplementary Fig 6d, e). In the latter case, the differences in stiffness and storage

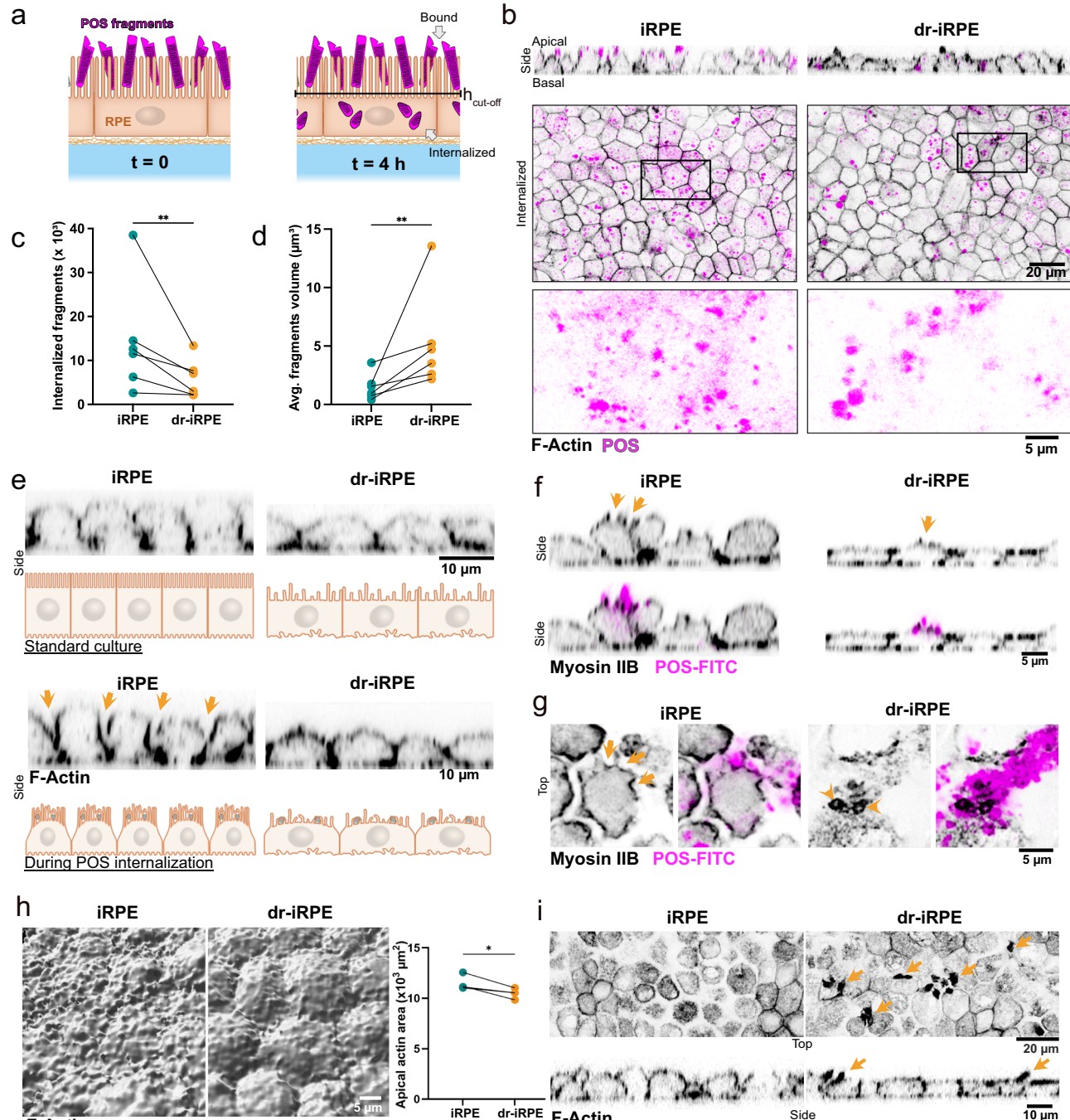

**Fig. 2 | Density reduced iRPE shows altered internalisation of POS fragments.**
**a** Schematics of POS fragment internalisation assay. Isolated and labelled porcine POS fragments are fed to iRPE together with bridging molecules at timepoint 0 and analysed four hours later. To distinguish between bound and internalised fragments, we define a cut-off height. **b** Representative orthogonal cross-sections and single optical slices of iRPE and dr-iRPE at t = 4 h. Zoom-in of the POS signal from the inlet showcases the size of the fragments. **c** Quantification of fragments internalised within the time of the assay. Values are shown as times thousand per field of view (n = 6, p = 0.0079). **d** Quantification of the average volume of internalised fragments (n = 6, p = 0.0088). **e** Representative orthogonal cross-sections of F-actin labelled monolayers in a standard culture condition vs during the POS internalisation assay. Images are not paired. Schematic illustration and arrows highlight the distinct cross-sectional morphology of cells, with control cell bodies bulging out above the level of cell–cell junctions. **f** Representative orthogonal cross-sections of Myosin IIB labelled monolayers and FITC-labelled POS fragments. Myosin IIB is enriched in proximity to the POS (arrows). **g** Representative apical view of iRPE cells in control and density reduced condition, stained for Myosin IIB

during internalisation of POS-FITC. In contrast to many small phagocytic cups localised apically and laterally in iRPE (arrows), dr-iRPE shows fewer apical-only cup-like structures with more localised myosin signal (arrowheads).
**h** Representative 3D segmentation of apical F-actin surface of monolayers during internalisation assay (left) and quantification of apical actin area within the same field of view (right, n = 4, p = 0.0318). Datapoints in **c**, **d** and **h** represent average values per independent biological replicate, with n defined as an independently prepared monolayer, obtained from the same iRPE batch and cultured and treated in parallel. Statistical significance was tested using a two-sided ratio paired t-test, where *: p < 0.05 and **: p < 0.01. **i** Representative apical and orthogonal cross sections of F-actin-labelled monolayers during internalisation. Arrows indicate the presence of protruding actin-rich structures in dr-iRPE. Representative micrographs in **e**–**g** and **i** are representative of three independent experiments with similar results. Source data are provided as a Source Data file. iRPE: induced pluripotent stem cell-derived retinal pigment epithelium; dr-iRPE: density-reduced iRPE, POS: photoreceptor outer segments.

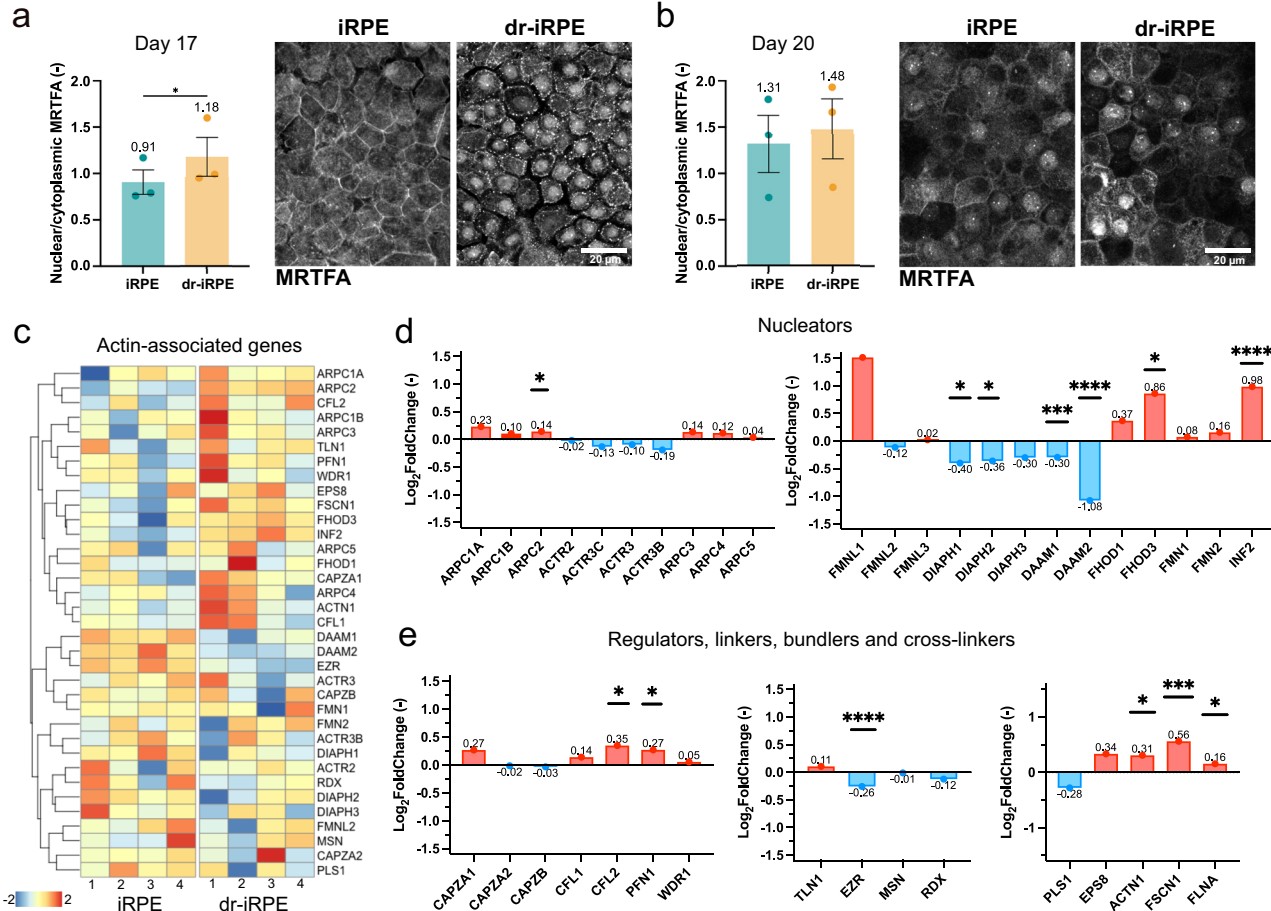

**Fig. 3 | A different fingerprint of actin-associated proteins in density-reduced iRPE. a** Quantification of nuclear to cytoplasmic MRTFA fluorescence intensities and representative micrographs of MRTFA staining one day after apoptosis induction (day 17, $n = 3$, $p = 0.0133$). **b** Quantification of MRTFA fluorescence intensities and representative micrographs of MRTFA staining on the day of readout (day 20, $n = 3$, $p = 0.3899$). Datapoints in **a** and **b** represent average values per experiment $\pm$ SEM originating from $n = 3$ independent biological replicates. Statistical significance was tested using a two-sided ratio paired t-test, where *: $p < 0.05$. **c** Hierarchically clustered heatmap of RNA transcript counts of actin-related proteins from $n = 4$ independent biological replicates of iRPE vs dr-iRPE samples. **d** Graphs showing the Log2Fold change in RNA transcript expression of branched (left) and linear (right) actin nucleator proteins for dr-iRPE vs. iRPE ($n = 4$, $p = 0.0440$ (ARPC2); 0.0565 (DIAPH1); 0.1013 (DIAPH2); 0.0004 (DAAM1);

$8 \times 10^{-15}$ (DAAM2); 0.0377 (FHOD3); $2 \times 10^{-5}$ (INF2)). **e** Graphs showing the Log2Fold change in RNA transcript expression of actin assembly/disassembly regulators (left), actin membrane linkers (middle) and crosslinkers and bundlers (right) for dr-iRPE vs. iRPE ($n = 4$, $p = 0.0454$ (CFL2); 0.0905 (PFN1); 0.0001 (EZR); 0.0160 (ACTN1); 0.0004 (FSCN1); 0.0867 (FLNA)). Differentially expressed genes in **d** and **e** are marked with *: $p_{adj} < 0.1$, **: $p_{adj} < 0.01$, ***: $p_{adj} < 0.001$, and ****: $p_{adj} < 0.0001$ as a result of the use of negative binomial generalised linear models and Wald test. n in all panels is defined as an independently prepared monolayer, obtained from the same iRPE batch and cultured and treated in parallel. Source data are provided as a Source Data file. iRPE: induced pluripotent stem cell-derived retinal pigment epithelium; dr-iRPE: density-reduced iRPE; MRTFA: myocardin-related transcription factor A.

modulus were smaller and only apparent at the lower probed frequencies (1 and 2 Hz). Lastly, medium transfer controls seemed to experience softening rather than stiffening, excluding an effect of apoptotic debris on the observed stiffening in dr-iRPE (Supplementary Fig. 6f). Collectively, these measurements revealed a reinforced cellular interconnectivity that results in a less deformable monolayer, i.e. stiffer, as well as more elastic, suggesting an actomyosin-based reinforcement[55].

As mechanical properties and contractility are often linked[56,57], we examined the levels of phosphorylated myosin light chain 2 (pMLC2) to obtain a general idea about actomyosin contractility[58]. Immunoblotting revealed general increased levels of pMLC2 in dr-iRPE (Supplementary Fig. 6c), pointing to overall higher actomyosin contractility, especially localised at the cell-cell junctions as shown by immunofluorescent staining (Fig. 4d and d[I]). This observation was further supported by enrichment of vinculin at the cell-cell and at the tricellular junctions (Fig. 4e, e[I]), pointing to higher contractile force at cell-cell junctions[59].

Nanoindentation enabled us to characterise the biomechanical properties of the RPE from the apical side. Our indentation setup was designed to probe multicellular length scale deformation, where the indentation of the 20 µm tip at the selected depth is counteracted by intercellular junctions[60] (Fig. 4a, Supplementary Fig. 6a, Supplementary Movie 2). To characterise subcellular mechanical properties of the RPE monolayer non-invasively, i.e., independently from intercellular adhesion, we applied Brillouin micro-spectroscopy. Here, the detection of photons, inelastically scattered by thermally-induced acoustic phonons, enables the probing of viscoelastic properties within the sample at microscale resolution[61]. Using Brillouin microscopy, we first performed low-resolution z-scans through iRPE and dr-iRPE monolayers, including the surrounding hydrogel and medium (Fig. 4f). These line profiles revealed the presence of a peak in the Brillouin shift, corresponding to the cellular basal side, which gradually decreased towards the apical cell region (Fig. 4g). The quantification of fitted peak values of these line profiles revealed significantly lower Brillouin shift in dr-iRPE compared to iRPE (Fig. 4h). Moreover, based on the

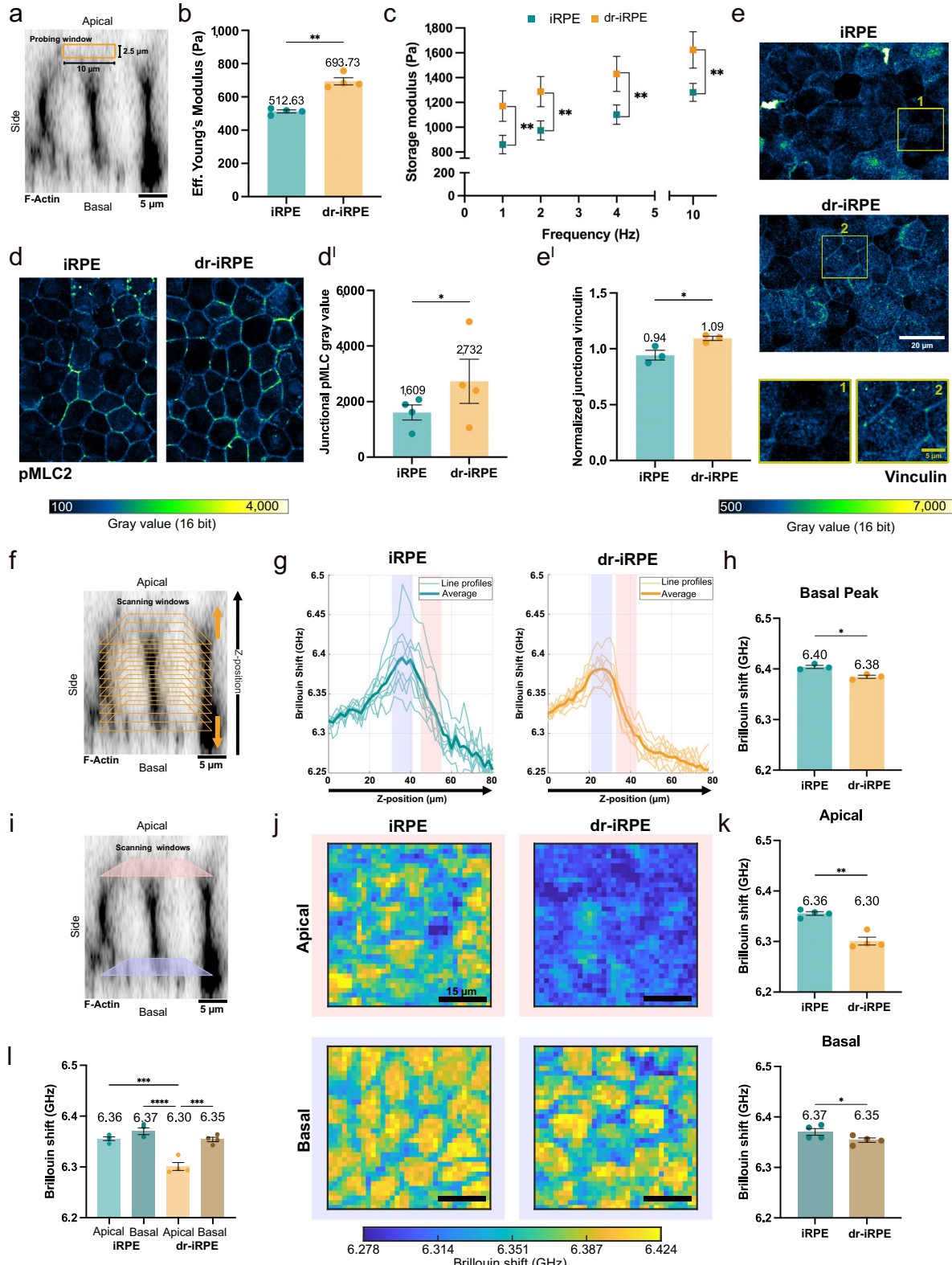

peaks shown in Fig. 4g and confirmed by live-cell brightfield imaging, we identified the apical and basal z-levels for conducting 2D high resolution raster scans (Fig. 4i). The 2D raster scans (50 x 50 µm) revealed mechanical compartmentalisation into areas with different Brillouin shifts, which may correspond to cellular and intercellular regions (Fig. 4j). Notably, this compartmentalisation was much more evident at the basal sides. The quantification of the mean Brillouin

shifts revealed a difference between the conditions, with dr-iRPE cells showing lower average values at the apical (6.30 vs 6.36 GHz) and basal (6.35 vs 6.37 GHz) sides of the cells (Fig. 4k). Interestingly, when comparing apical and basal shifts within the same conditions, dr-iRPE cells showed significantly different apical shifts compared to basal, which was not the case in iRPE (Fig. 4l). As the Brillouin shift reflects cytoskeletal organisation and water content[61–63], this supports the idea

**Fig. 4 | Density reduced iRPE presents a new biomechanical state. a** Image illustrating the probing window (10 ×2.5 μm) during nanoindentation on an orthogonal cross-section of F-Actin-labelled live iRPE. **b** Average Effective Young's Modulus measured by nanoindentation scans in iRPE and dr-iRPE (*n* = 4, *p* = 0.0026). **c** Average storage modulus ± SEM at different oscillation frequencies (*n* = 4, *p* = 0.0035 (1 Hz); 0.0040 (2 Hz); 0.0056 (4 Hz); 0.0079 (10 Hz)). **d** Representative images of pMLC2-labelled iRPE monolayers with grey values represented in the colour bar. **dᴵ** Quantification of junctional pMLC2 signal (*n* = 4, *p* = 0.0490). **e** Representative images of vinculin-labelled iRPE monolayers with annotated regions enlarged. **eᴵ** Quantification of junctional fluorescence signal of vinculin normalised to cytoplasmic fluorescence signal (*n* = 3, *p* = 0.0378). **f** Image illustrating the geometry of low-resolution Brillouin z-scans on an orthogonal cross-section of F-Actin-labelled live iRPE. **g** Representative line profiles of the z scans showing the Brillouin shift values starting from the hydrogel, across the iRPE monolayers and reaching into the cell medium. Averages of z-scans are represented with thicker lines. Highlighted in light lilac and salmon are the approximate basal and apical cell regions, respectively. **h** Quantification of Brillouin shift values obtained by fitting the peaks of the curves shown in **g** (*n* = 3, *p* = 0.0124). **i** Image illustrating high-resolution 2D raster scan windows on an orthogonal cross-section of F-Actin-labelled live iRPE. **j** Representative Brillouin shift maps obtained from the 50 ×50 μm apical and basal scans of the iRPE monolayers. **k** Quantification of mean Brillouin shift values from 50 ×50 μm regions of interest recorded in apical and basal regions in iRPE monolayers (*n* = 4, *p* = 0.0032 (apical); 0.0347 (basal)). **l** Comparative bar plot of apical and basal Brillouin shifts in iRPE and dr-iRPE cells (*n* = 4, *p* = 0.0001 (Apical-iRPE vs Apical-dr-iRPE); <0.0001 (Basal-iRPE vs Apical-dr-iRPE); 0.0002 (Apical vs Basal dr-iRPE)). Data points in **b, dᴵ, eᴵ, h, k** and **l** represent average values per independent biological replicate ± SEM. In all panels, n is defined as independently prepared monolayer, obtained from the same iRPE batch and cultured and treated in parallel. Statistical significance in **b** and **c** was tested using a two-sided paired t test, in **dᴵ** and **eᴵ**—using a two-sided ratio paired t test, where *: p < 0.05, **: p < 0.01, ***: p < 0.001 and ****: p < 0.0001. Statistical significance in **h** and **k** was tested using a two-sided paired t-test, and in **l** using a one-way ANOVA, where *: p < 0.05, **: p < 0.01, ***: p < 0.001, and ****: p < 0.0001. The micrograph in **a, f** and **i** is representative of three independent experiments with similar results. Source data are provided as a Source Data file. iRPE: induced pluripotent stem cell-derived retinal pigment epithelium; dr-iRPE: density-reduced iRPE; pMLC2: phosphorylated myosin light chain 2.

that the apical sides of the cells undergo the most pronounced reorganisation in dr-iRPE. This is consistent with microvilli and apical actin remodelling observed in dr-iRPE, i.e., transcellular stress fibres and F-actin surface topography.

As the intercellular junctions showed an increased pMLC2, thereby an increased contractility and stiffness, we sought to isolate the shifts of "cell edges" from those of "cell bodies". As this distinction was not clear in the apical scans, we performed the analysis on the basal sides, where the distribution of shift values revealed two distinct modes (Supplementary Fig. 7a, b). Quantification based on segmented Brillouin shift plots that, regardless of the experimental condition, the interstitial regions of the monolayer ("cell edges") consistently exhibit lower shifts than the cell cores ("cell bodies"). This also confirms a trend toward reduced core shifts in dr-iRPE independently of the interstitium (Supplementary Fig. 7c, d). Notably, the shift quantified at the interstitium was unexpectedly lower than at the cell bodies, despite enrichment of F-actin. This likely reflects the absence of well-defined intercellular junctions at the basal side, and instead, the presence of loosely interdigitated basal membranes visible with F-actin staining (Supplementary Fig. 7e). These looser basal intercellular interactions likely explain why structural and mechanical remodelling following cell density reduction is predominantly apical, affecting load-bearing apical junctions rather than the more compliant basal domain.

Altogether, biomechanical characterisation at the two different length scales showed that, in addition to the structural and functional adaptation, the reduction of cell density causes RPE to adopt a different mechanical state. This mechanical state is associated with increased contractility at the apical intercellular junctions, which confers a stiffer and more elastic tissue, as evident by the nanoindentation. Brillouin shift analyses at the subcellular level also confirmed distinct mechanical states between the two conditions, most pronounced at the apical surface. Taken together, this apical prevalence of the mechanical changes may result from the intimate link between the epithelial junctional complex and the organisation of the apical domains, which in the case of the RPE may create a mechanical feedback loop between tissue integrity and function.

### Actin modulation differentially impacts POS phagocytosis

Motivated by our observations on cortical actin topography, mechanics, and the differential expression of actin-related regulators, we aimed to directly investigate the cause–effect relationship between POS internalisation and actin remodelling. To this end, we targeted actin nucleation, a key process influencing cortical thickness and membrane tension[64]. For the experimental design, we considered the upregulation of Arp2/3 subunit mRNA expression and the downregulation of several formins in dr-iRPE (Fig. 3d), along with the distinct actin structures observed during internalisation. Dr-iRPE exhibited a larger, flatter actin surface topography (Fig. 2h), suggesting an abundance of heavily branched actin, similar to that found in lamellipodia. In contrast, control samples displayed more numerous, smaller, and protruding actin structures (Fig. 2h), resembling filopodia, which are characterised by linear actin cores and highly dependent on formin activity[65]. Based on these findings, we inhibited Arp2/3 activity using the compound CK-666[66] and inhibited formin activity using the compound SMIFH2[67].

Formin inhibition in iRPE did not affect the cell height but significantly stiffened the monolayer on a tissue level (Supplementary. Fig. 8a, b). In dr-iRPE, the inhibition did not produce a change in cell height, but led to a softer monolayer (Supplementary Fig. 8c, d). When tested for the POS fragments internalisation, the quantification showed a highly variable effect on the number of internalised particles, which nevertheless showed an overall significant reduction in the number of both internalised and bound POS fragments (Fig. 5a, b; Supplementary Fig. 8e and f). In contrast, no difference was detected in SMIFH2-treated dr-iRPE (Supplementary Fig. 8g). Next, we quantified the fragments' size, which was significantly affected by SMIFH2 only in iRPE, resulting in larger fragments on average (Fig. 5a, b, Supplementary Fig. 8h). Since SMIFH2 affected internalisation of POS only in iRPE, we further analysed 3D-segmented actin and monolayer deformation during internalisation. SMIFH2-treated iRPE monolayers displayed a smoother apical actin topography; however, this was not captured by the quantification of the total apical actin area (Fig. 5c; Supplementary Fig. 8i). Furthermore, SMIFH2-treated iRPE monolayers also bulged out upwards similarly to untreated controls, but showed denser F-actin signal at the junction, suggesting a lateral reinforcement comparable to dr-iRPE (Fig. 5d). In addition, the bulging cells showed a more irregular apical morphology (Fig. 5d).

Although SMIFH2 is widely accepted as a reliable formin inhibitor, it has been reported to have a concentration-dependent off-target effect on Myosin II activity[68]. To evaluate whether the effects observed with 20 μM SMIFH2 were not due to partial Myosin II inhibition, we conducted SMIFH2 treatments across a range of concentrations (5–100 μM). We then assessed iRPE and dr-iRPE monolayers for changes in stiffness and phagocytic activity (Supplementary Fig. 8j, l, n, p, r, t). As a positive control for Myosin II inhibition, we used blebbistatin (Supplementary Fig. 8k, m, o, q, s, u). In iRPE, stiffness quantification confirmed a significant monolayer stiffening with 20 μM SMIFH2, whereas blebbistatin treatment induced an expected softening as cellular contractility is reduced (Supplementary Fig. 8j, k). Differently,

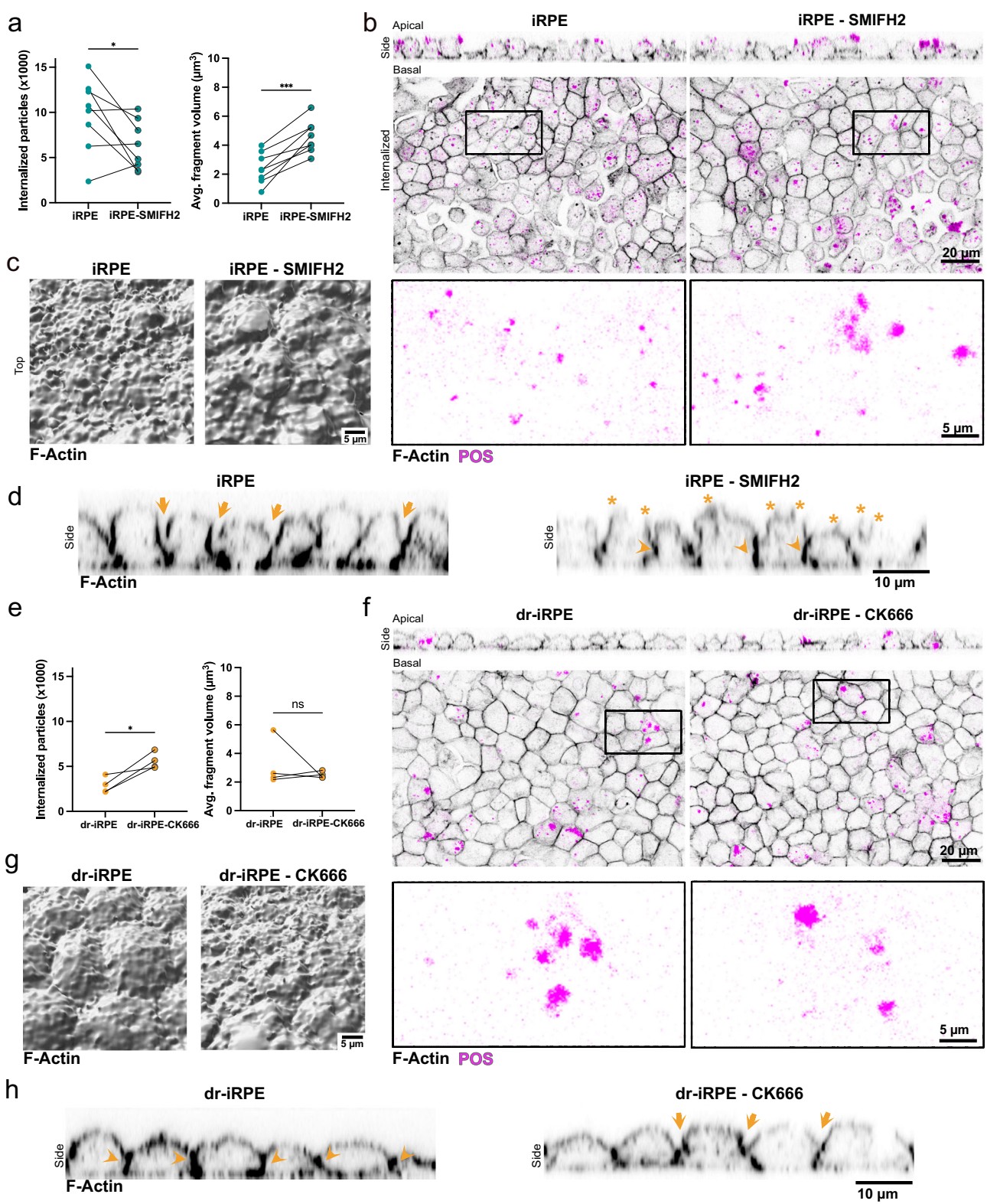

dr-iRPE treated with SMIFH2 displayed an overall softening effect, most prominent at 100 µM, consistent with the trend observed following blebbistatin treatment (Supplementary Fig. 8l, m). Finally, analyses of the POS internalisation process revealed that both SMIFH2 at all concentrations and blebbistatin reduced the number of internalised POS particles in iRPE. Still, only the low concentrations of SMIFH2 (5 and 20 µM) led to increased average phagosome size, while blebbistatin did not affect fragment size (Supplementary Fig. 8n–q).

In dr-iRPE, SMIFH2 significantly reduced the number of internalised particles only at 5 µM with no effect at higher concentrations, whereas blebbistatin reduced it (Supplementary Fig. 8r, s). Average phagosome size was clearly reduced by SMIFH2 treatment in dr-iRPE, in opposite direction to iRPE (Supplementary Fig. 8t), while blebbistatin did not affect the phagosome size (Supplementary Fig. 8u). Overall, while we cannot fully exclude the possibility of minor off-target inhibition of Myosin II by 20 µM SMIFH2, we are confident that

**Fig. 5 | Actin modulation via nucleator inhibitors affects internalisation of photoreceptor outer segments. a** Number of internalised fragments (left, *p* = 0.0424) and their average volume (right, *p* = 0.0004) for iRPE vs iRPE treated with 20 μM SMIFH2 (*n* = 8). **b** Representative orthogonal cross-sections and single optical slices of iRPE and SMIFH2-treated iRPE at t = 4 h. Zoom-in of the POS signal from the inlet showcases the size of the fragments. **c** Representative 3D segmented apical F-actin surface of monolayers during POS internalisation assay for iRPE vs SMIFH2-treated iRPE. **d** Representative orthogonal cross-sections of F-actin labelled monolayers during POS internalisation assay of untreated and SMIFH2-treated iRPE. Arrows indicate the junctional remodelling to induce cell bulging, arrowheads indicate the junctional reinforcement, and stars highlight apical irregularities. **e** Number of internalised fragments (left, *p* = 0.0291) and their average volume (right, *p* = 0.4919) for dr-iRPE vs dr-iRPE treated with 500 nM CK-666 (*n* = 4). **f** Representative orthogonal cross-sections and single optical slices of dr-iRPE and CK666-treated dr-iRPE at t = 4 h. Zoom-in of the POS signal from the inlet

showcases the size of the fragments **g** 3D segmented apical F-actin surface of monolayers during POS internalisation assay for untreated vs CK666-treated dr-iRPE. **h** Representative orthogonal cross-sections of F-actin labelled monolayers during photoreceptor fragment internalisation assay of untreated and CK666-treated dr-iRPE. Arrows indicate the junctional remodelling to induce cell bulging, while arrowheads indicate the junctional reinforcement. Datapoints represent average values per independent biological replicate with n defined as an independently prepared monolayer, obtained from the same iRPE batch and cultured and treated in parallel. Statistical significance in **a** and **e** was tested using a two-sided ratio paired t-test, where *: *p* < 0.05. Representative micrographs in **b–d** and in **f–h** are representative of three independent experiments with similar results. Source data are provided as a Source Data file. iRPE: induced pluripotent stem cell-derived retinal pigment epithelium; dr-iRPE: density-reduced iRPE; POS: photoreceptor outer segments; SMIFH2: formin inhibitor; CK-666: Arp2/3 inhibitor.

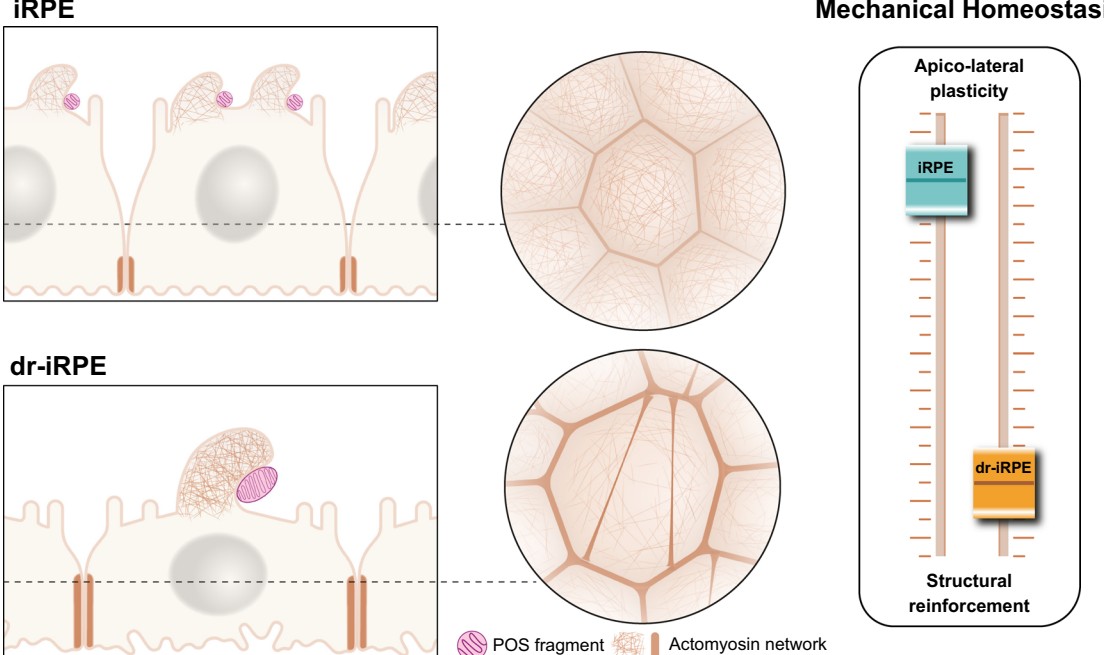

**Fig. 6 | Cell density reduction leads to a shift in RPE mechanical homeostasis.** The density reduction in iRPE leads to structural remodelling and biomechanical alterations that recapitulate the tissue architecture of the aged native RPE, suggesting a shift in mechanical homeostasis. These changes correlate with a reduced number of internalised POS fragments, which are larger in size, highlighting functional decline. Actin cytoskeleton remodelling mediates structural reinforcement and functional alteration, suggesting a trade-off where tissue structural preservation compromises the apico-lateral plasticity essential for phagosome formation and POS phagocytosis. iRPE: induced pluripotent stem cell-derived retinal pigment epithelium; dr-iRPE: density-reduced iRPE; POS: photoreceptor outer segments.

Myosin II inhibition does not drive our results. This confidence stems from the clear differences in trends for iRPE and dr-iRPE regarding both stiffness and internalised POS number and volume, as well as the contrasting effects seen with blebbistatin treatments.

The inhibition of Arp2/3 using CK-666 in iRPE did not change the cell height or stiffness, whereas in dr-iRPE, the treatment led to a slightly increased cell height (Supplementary Fig. 9a, b), but no significant changes in monolayer stiffness (Supplementary Fig. 9c, d). During phagocytosis, no effect was detected when iRPE were treated, but in dr-iRPE, the treatment led to an increased number of internalised particles (Fig. 5e, f; Supplementary Fig. 9e, g), while the average internalised particle size and the number of bound particles remained comparable (Fig. 5e, g, Supplementary Fig. 9f, h). As the phagocytic process was affected only in dr-iRPE, we proceeded to analyse the morphology of the 3D-segmented cortical actin during internalisation. The analyses showed the occasional presence of thinner and longer ruffles (Fig. 5g), which resulted in a significant increase in the quantified apical actin area (Supplementary Fig. 9i). Furthermore, orthogonally, the treated dr-iRPE showed some degree of bulging during

phagocytosis, which was more pronounced than in the untreated dr-iRPE (Fig. 5h).

Overall, these results demonstrate that modulating actin cytoskeleton remodelling by interfering with actin nucleation effectively altered cortical actin responsiveness during internalisation. These manipulations were sufficient to replicate or rescue differences in POS fragments internalisation observed in dr-iRPE. Particularly, Arp2/3 activity appears to be related to cell bulging, modulating apical actin area and cell height, which impacts the number of internalised particles, while formins activity modulates tissue-level stiffness, affecting both the number and size of the internalised particles.

These insights strongly indicate that dr-iRPE cells establish a new mechanical equilibrium, in which the actin cytoskeleton acts as a critical link between tissue structure, mechanics, and the execution of phagocytosis. The altered organisation of the actomyosin network enhances contractility at cell–cell junctions, and simultaneously reduces the capacity for apical remodelling required for the formation of phagosomes. We therefore conclude that, in response to reduced cell density, the RPE monolayer reroutes actin-related processes

toward reinforcing tissue integrity. This shift diminishes the plasticity of the actin-myosin network, which is essential for efficient POS phagocytosis (Fig. 6).

## Discussion

In this work, we present an in vitro model that recapitulates structural and functional features of the aged RPE. The model is inspired by the natural loss of cells through apoptosis within the native RPE during ageing. Comparing dr-iRPE with iRPE, we demonstrate significant changes in biomechanical features that evidence a shift in RPE mechanical equilibrium related to the structural remodelling to accommodate the reduced cell density. Most importantly, this shift has implications for the function of POS internalisation, potentially leading to impaired tissue performance. We identify remodelling of the actin cytoskeleton as a key factor mediating both the structural tissue adaptation to cell loss and the observed functional differences, suggesting that tissue preservation may be prioritised at the expense of functional trade-offs in ageing RPE. Ultimately, this structural adaptation disrupts the actomyosin machinery's ability to effectively remodel, thereby compromising efficient phagosome formation and POS phagocytosis (Fig. 6).

Ageing is a multifactorial phenomenon that impairs the capacity for dynamic homeostatic adjustments[69]. Only recently has the scientific community acknowledged altered mechanical properties as a hallmark of ageing[70], raising interest in mechanobiological studies in ageing. With the aim of studying the effect of reduced cell density on the structural and functional adaptation of RPE, our model highlights the importance of tissue organisation and mechanical properties in ageing. By using epigenetically young[71] cells derived from human induced pluripotent stem cells (hiPSCs), which have undergone rejuvenation-associated reprogramming, we reduced the complexity down to examining how tissue architecture alone impacts mechanical homeostasis and functional capacity, putting aside more complex aspects of biochemical signalling such as oxidative stress. In this system, we were able to recapitulate structural aspects identical to reported remodelling of the aged RPE, such as cell height reduction[31], shortening of the apical microvilli[72] and presence of actin stress fibres[33] by only reducing cell density. This remodelling is possible due to the nearly postmitotic nature of iRPE cells, which cannot use compensatory proliferation as a homeostatic mechanism, but are believed to rely on compensatory cellular hypertrophy[73,74]. We noted that cellular hypertrophy in our model is not merely cell volume enlargement, but tissue stretching from within. We observed cell area enlargement paired with cell height reduction, pointing to individual cell volume conservation, which may enrich the current notion of compensatory hypertrophy in postmitotic epithelia[74]. Individual cell area enlargement could also explain the observed microvilli shortening, as microvilli provide membrane reservoirs capable of unfolding to facilitate cell shape changes or to buffer mechanical stress[75,76]. In light of the cell density changes in RPE pathologies such as proliferative vitreoretinopathy and age-related macular degeneration[77,78], future studies should explore the relationship between the degree of cell loss and the structural, functional or cell-cycle-related adaptations.

The changes induced by cell density reduction are also paired with a shift in the range of mechanical equilibrium. This shift is evidenced by multicellular length-scale stiffening and increased elasticity, with softening at the apical side at the subcellular length scale. Tissue stiffening, in particular increased elasticity, has been reported as a response of epithelial cells to stretch[75,79], at least in the short term. In our system, the mechanical changes are observed in the long term, four days after the apoptosis-mediated deformation. While these data reveal how RPE cells in vitro compensate for large-scale cell loss, the situation in vivo is undoubtedly more complex to decipher. Multiple factors influence native tissue during ageing, making a direct comparison between our reductionistic model and physiological RPE

ageing premature. Nevertheless, alterations in actin cytoskeletal organisation have been reported in aged RPE[33], supporting the notion that changes in mechanical homeostasis, including tissue stiffening, also occur in vivo. Such adaptation may arise from reduced cell density and the resulting need to reinforce tissue integrity. This is particularly evident in the formation of transcellular actin stress fibres, observed both in our model and in aged native tissue[33]. In fact, the occurrence of transcellular actin stress has been reported as a sign of junctional remodelling[80] and limiting cell size or elongation during morphogenesis[81]. Moreover, the strengthening of cell–cell junctions is a known mechanism to support cells in resisting forces[82]. Interestingly, the non-invasive probing technique of Brillouin microscopy revealed a mismatch between tissue reinforcement and apical cortex remodelling. Although the results of indentation measurements cannot be directly compared to light-based measurements[61,62,83], the observed lower Brillouin shift may be explained by significant changes in the apical cortex composition and in the apical microvilli, which are rich in actin and tubulin filaments. These filaments enhance the Brillouin shift due to their structural stiffness compared to the underlying cell[63]. A more elaborate apical cortex may also possess a different level of hydration, which has recently been shown to be a crucial factor affecting Brillouin, but not indentation-based measurements[62]. Furthermore, differential remodelling of the apical and lateral cortical actin networks is likely, as is known to happen during morphogenesis[84]. Notably, the quantified reduction of phosphorylated ezrin indicates decreased cortical tension in dr-iRPE, consistent with both our Brillouin measurements and existing literature[85]. Finally, we were surprised to find that many structural and mechanical adaptations are associated mainly with the apicolateral domains. In fact, a direct interplay between the structural and mechanical remodelling of cell–cell and cell–ECM adhesions has been widely reported[86–91]. Except for epithelial buckling during morphogenesis[88] or pathology[92], this homeostatic balance is generally linear: reinforcement and increased tension at intercellular junctions are typically accompanied by enhanced cell–ECM adhesion and increased traction forces[87]. Nevertheless, in our case, no apparent reinforcement has been detected at the basal side of the monolayer. This may be attributed to our study conditions, where iRPE cells are cultured on polyacrylamide hydrogels, which provide strictly controlled physical and biochemical cues with a limited possibility for cell-mediated remodelling. Future studies need to elucidate the contribution of ECM cues and cell-mediated ECM remodelling as part of the puzzle of age-related structural remodelling of the outer retina.

Our work suggests that age-associated structural phenotype goes hand in hand with actomyosin organisation and remodelling capacity, which affects the RPE function of POS fragments internalisation. Phenotypically, we observe an overall higher contractility and shifts in the expression of nucleators and other actin-related proteins. The apical domain of RPE cells requires high remodelling capacity due to its essentiality for the internalisation function[49]. In this study, we observed an apicobasal cell remodelling with bulging and lateral relocalisation of cell-cell junctions during the POS internalisation assay, which has not been reported before. Extensive studies on phagocytosis in macrophages have demonstrated that efficient phagocytic cup formation requires coordinated mechanical remodelling of the plasma membrane and cortical actin[93–95]. In contrast to isolated cells, RPE cells cannot remodel their apical membrane without simultaneously remodelling the interaction with their neighbours, due to their tight epithelial organisation. Therefore, unlike single-cell phagocytosis, phagocytosis in the RPE can be envisioned as a more collective phenomenon, in which neighbouring cells coordinate lateral remodelling to support apical bulging and phagosome formation. The observed phenomena may arise from the synergy of conditions in our culture system, where we observe mature cells of nearly three weeks cultured on soft substrates, as opposed to the majority of other

studies, which are being conducted on stiff cell culture plastics or glass substrates. In our model of cell density reduction, we observed a lack of apical bulging and prominent actin-rich protrusions during internalisation. The absence of bulging may stem from the difficulty in disassembling junction-associated structures, which are more reinforced and stable, or from their connection to highly crosslinked or bundled actin structures. A logical explanation is that increased contractility at the cell-cell junction, necessary to maintain tissue integrity, may redirect actin dynamics from the apical side. This could limit the cytoskeletal and membrane flexibility required for efficient apical processes, including phagosome formation and the development of sufficient tension to engulf POS fragments. The observed formation of large actin-rich protrusions exclusively in dr-iRPE is reminiscent of arms structures described in the early embryo[96] and may reflect an altered remodelling capacity, or plasticity, of the membrane-cortex complex. The formation of these protrusions is driven by Arp2/3 activity[96], which is highly versatile both in maintaining junctional integrity[97], regulating cortical thickness and tension[64], as well as playing a major role in the internalisation process[48,50]. Its activity may be diverted towards a new housekeeping maintenance of cortical and junctional actin structure, as opposed to on-demand availability for internalisation. The role of formins in RPE function remains unexplored to date, but our data suggest that they both influence the efficiency of POS phagocytosis and phagocytic cup size when inhibited. Interestingly, formin inhibition in RPE led to monolayer stiffening, opposite to studies examining single, sub-confluent or non-polarised cells[98,99]. This possibly reflects a response of tension redistribution occurring in mechanically integrated, static monolayers like the postmitotic RPE, where cell-cell adhesions are in mechanical equilibrium. The effect on phagocytosis may occur through alterations in membrane curvature or surface ruffling, impacting the contact area for internalisation. Notably, inhibition of Myosin II with blebbistatin did not affect the size of internalised POS fragments, but only their number, thereby excluding a role for contractility in regulating phagosome size.

Our findings demonstrate that structural adaptation to cell loss reprogrammes actin organisation, stabilising filaments and redirecting actin towards lateral junctions. This reinforcement limits cytoskeletal responsiveness at the apical surface, constraining the dynamic remodelling required for POS phagocytosis. Thus, the monolayer adopts a new mechanical homeostasis that prioritises structural integrity over functional plasticity. We propose that this stability–flexibility trade-off is a hallmark of postmitotic tissues, offering protection at the expense of specialised functions. This mechanobiological shift provides insight into ageing-related decline in tissues such as the retina and brain[24,25,100–103], and may extend to proliferative epithelia. Future work should determine how density and architecture-driven changes in mechanics contribute to progressive functional loss across tissues.

## Methods

### Preparation of cell hydrogel substrates
The surface of 20 mm glass-bottom dishes (Cellvis, D35-20-0-N) or 6-well glass bottom plates (Cellvis, P06-1.5H-N) was activated to allow hydrogel attachment by etching with 0.1 M NaOH (Merck, 106498), followed by reaction with 4% (3-Aminopropyl)triethoxysilane (Sigma Aldrich, 440140) in isopropanol (VWR, 20842) and final incubation with 1% Glutaraldehyde (Merck, 354400) in double distilled water (ddH2O) (B. Braun, 0082479E).

Polyacrylamide hydrogels of 4 kPa stiffness were prepared from an 5%/ 0.1% Acrylamide/Bis-Acrylamide (Bio-Rad 1610140, Bio-Rad 1610142) mixture in ddH2O by radical polymerisation upon addition of the initiators ammonium persulfate (Sigma Aldrich, 248614) at 0.5% (vol/vol) and N'-Tetramethylethylenediamine (Sigma Aldrich, 411019) at 0.05% (vol/vol)[104].

Hydrogels were trimmed to size of 4 mm before functionalisation with Laminin-511 (Biolamina, LN511) and Laminin-332 (Biolamina, LN332) at a concentration of 15 μg/ml respectively. Briefly, a functionalisation solution was prepared from ddH2O, 0.5 M HEPES (Carl Roth, 9105) pH 6, 100% ethanol (Sigma Aldrich, 32205) and 0.2% (vol/vol) Bis-Acrylamide. The solution was degassed for 20 min and was completed by adding 0.2% Di(trimethylopropane)tetraacrylate (Sigma Aldrich, 408360), 3% (wt/vol) Hydroxy-4'-(2-hydroxyethoxy)-2-methylpropiophenone (Sigma, 41089 and 0.03% (wt/vol) Acrylic acid N-hydroxysuccinimide ester (Sigma, A8060)[105]. Hydrogels were dehydrated with 70% ethanol for 5 min, incubated under UV for 10 min with the functionalisation solution and washed with 25 mM HEPES and PBS for a total of 20 min before adding the protein solution. The gels were incubated with the protein solution overnight at 4 °C.

### Culture of human-induced pluripotent stem-cell-derived RPE
HiPSCs-derived RPE cells (Fujifilm Cellular Dynamics, R1102) were seeded in 24-well plates coated with 2.5 μg/ml vitronectin XF (StemCell Technologies, 07180). Cells were cultured in MEM-alpha with nucleosides (Thermo Fisher, 12571063), supplemented with 1x Glutamax (Thermo Fisher, 35050061), 5% (vol/vol) Knockout Serum replacement (Gibco, 10828028), 1% (vol/vol) N2-Supplement (Gibco, 17502048), 55 nM Hydrocortisone (Sigma Aldrich, H6009), 250 μg/ml Taurine (Sigma Aldrich, T8691), 14 pg/ml Triiodo-L-thyronine (Sigma-Aldrich, T5516) and 25 μg/ml gentamycin (Gibco, 15750060) following the manufacturer's instructions. The medium was exchanged every other day. For seeding on hydrogels, cells were detached from the 24-well plate by 30 min incubation with 5 mM PBS/EDTA (Sigma-Aldrich, E6511), followed by 5 min incubation with TrypLE Express (Gibco,126050). Cells were seeded on the UV-sterilised functionalised hydrogels at 5000–6000 cells/mm$^2$. The medium of cells on hydrogels was exchanged halfway twice a week.

### On-demand apoptosis for generation of density-reduced iRPE (dr-iRPE)
Apoptosis was induced by expression and on-demand dimerisation of recombinant FKBP-caspase8 (FKBP-casp8) fusion protein[41]. The FKBP-casp8 containing plasmid was delivered packaged in AAV5 (Vector-Builder, VB230127-1043eaq). Briefly, on day 10 after seeding on the hydrogel, cells were transduced with the virus at $7 \times 10^6$ infectious units per cell for 24 h. On day 15, 120 h after the transduction, FKBP-casp8 was dimerised by addition of 100 nM AP20187 in medium (Sigma Aldrich, SML2838) for 24 h. Paired control monolayers were also subjected to the AP20187 treatment. The read-out was performed on day 20, or 96 h after apoptosis treatment finalisation.

### Virus, mixed and medium-transfer controls
To exclude effects of the AAV as a delivery system, we used an AAV5-packaged construct that promotes the expression of green fluorescent protein (GFP) (VectorBuilder, VB10000-9287ffw) as a control. Cells were transduced and treated the same way as described for FKBP-casp8 expressing cells, including same timing of viral transduction, AP20187 treatment and read-out.

To obtain a mixed population of transfected and naïve cells, 2 days prior to seeding on hydrogels, we transduced hiPSCs-derived RPE in a 24-well plate at $7 \times 10^6$ infectious units per cell for 24 h with the FKBP-casp8 vector (VectorBuilder, VB230127-1043eaq). The virus solution was replaced with fresh medium after 24 h. After another 24 h, we seeded a 10%/90% mixture of the transduced hiPSCs-derived RPE cells and naïve hiPSCs-derived RPE on the hydrogels as described above. On day 15, corresponding to the same apoptosis timing of dr-iRPE generation, apoptosis was induced by addition of 100 nM AP20187 (Sigma Aldrich, SML2838) for 24 h. The read-out was performed at 96 h after apoptosis treatment finalisation.

To exclude possible effects of the release of apoptotic debris during the apoptosis, 16 h after AP20187 addition, the medium from dr-iRPE samples was transferred onto control conditions. The medium was then replaced after apoptosis treatment finalisation. 24 h after apoptosis finalisation, the medium from dr-iRPE samples was again transferred to control samples and the samples were maintained in this conditioned medium until read-out.

### EdU incorporation-based proliferation assay

Proliferation was assessed by evaluating the incorporation of the base 5-Ethinyl-2'-Desoxyuridin (EdU) with the Click-iT EdU kit (Thermo Fisher, C10337). Cells were incubated with 1:1 mixture of 20 µg/ml EdU and cell culture supernatant for 24 h. The subsequent EdU development reaction was performed following the manufacturer's instructions. Monolayers were imaged completely as overviews of stitched tiles at inverted epi-fluorescent microscope Zeiss Axio Observer 7. Positive cells were identified by using the particle analysis tool in Fiji and their number was normalised to the total number of cells, as detected by DNA labelling by DAPI (1 µg/ml, Invitrogen D1306).

### Photoreceptor fragment phagocytosis assay

Photoreceptor outer segment fragments were isolated from porcine eyes and labelled with FITC (Thermo Fisher, F1906), following an established protocol[106]. For the assay, the vial was thawed and washed with medium at 2300 g for 5 min. The pellet was resuspended in complete hiPSCs-derived RPE medium together with the bridging molecules human Protein-S (Coachrom, pp012A) at 2 µg/ml and human MGF-8 (R&D Systems, 2767-MF-050) at 2.4 µg/ml. An approximate ratio of ten photoreceptor outer segment fragments per RPE cell seeded was used. Cells were incubated for 4 h, washed four times with DPBS with Ca$^{2+}$ and Mg$^{2+}$ (Gibco, 14040117) and fixed with 2% paraformaldehyde (Merck, 818715) for 8 min at room temperature (RT). Samples were then permeabilised with 0.3% Triton-X-100 (Sigma Aldrich, T8787) for 2 min and labelled for actin with Phalloidin AF568 (Molecular probes, A12380). Three z-stacks per monolayer were recorded at Zeiss LSM710 Confocal Microscope with ZEN Black 2.1 SP3 software with 40× water immersion objective (ZEISS, 420862-9970-000) in Airyscan super resolution mode. The subsequent analysis was performed in Imaris v10.2.0 (Oxford Instruments).

### Nanoindentation and dynamic mechanical analysis

Nanoindentation was performed on a Chiaro Nanoindentor (Optics 11 Life) mounted on an inverted epifluorescence microscope Zeiss Axio Observer 7 with environmental control. Measurements were performed in the Piuma V3.5.0 software on living cells in displacement control mode using a spherical probe of 20 µm diameter and 0.025 N/m stiffness. Three matrix scans of 16 indentations were recorded per monolayer, indenting 6 µm into the cells. Dynamic mechanical measurements were performed at 1, 2, 4 and 10 Hz to allow for viscoelastic characterisation. For the analysis, indentation portions between the contact point and 2.5 µm were fitted using the Hertz model in the Optics 11 Analysis Software DataViewer v2.6.0, allowing a range for the contact point fit of 30% of the maximum load. Effective Young's modulus values of fit $R^2$ better than 0.95 were used for further statistics. Loss and storage moduli were determined for the different frequencies (1, 2, 4 and 10 Hz) using the software default methods.

### Immunofluorescent labelling and imaging

Monolayers on hydrogels were fixed with 2% paraformaldehyde (Merck, 818715) for 8 min at RT and permeabilised with 0.3% Triton-X-100 (Sigma Aldrich, T8787) for 2 min at RT. Samples were then blocked with 1% BSA (Serva, 11930) in PBS for 30 min at RT, followed by incubation with primary antibodies in blocking solution overnight at 4 °C. The following primary antibodies were used: phospho-myosin light chain 2 (Ser19) (Cell Signalling Technology, Cat# 3671) at 1:50, vinculin

(Sigma-Aldrich, Cat# V4139) at 15 µg/ml, ezrin (Abcam, Cat# ab4069) at 10 µg/ml and non-muscle myosin heavy chain IIB (BioLegend, Cat# 909901) at 4 µg/ml. After washing, secondary antibodies diluted in PBS were added for 2 h at RT. Secondary antibodies included: Alexa Fluor 488 anti-mouse IgG (H + L) (Invitrogen, Cat# A11001) at 8 µg/ml, Alexa Fluor 594 anti-mouse IgG (H + L) (Invitrogen, Cat# A11005) at 8 µg/ml, Alexa Fluor 594 anti-rabbit IgG (H + L) (Invitrogen, Cat# A11012) at 8 µg/ml, Alexa Fluor 647 anti-rabbit IgG (H + L) (Jackson ImmunoResearch/Dianova, Cat# 111-605-144) at 7.5 µg/ml, Alexa Fluor 488 anti-rabbit IgG (H + L) (Jackson ImmunoResearch/Dianova, Cat# 711-546-152) at 8 µg/ml. Samples were washed and mounted in Mowiol mounting medium (Carl Roth, 9002-89-5) before imaging. Unless started otherwise, samples were imaged at Zeiss LSM710 Confocal Microscope with 40× water immersion objective (ZEISS, 420862-9970-000) in Airyscan super resolution mode as z-stacks of 0.5 µm optical slicing. Uniform imaging settings were used between samples within the same experiment. Images were then inspected and further processed for different analyses using Fiji[107].

### Bulk RNA sequencing

RNA isolation for RNA sequencing was performed using the RNeasy Micro RNA isolation kit (Qiagen, 74004) with DNase treatment following the manufacturers protocol. Bulk RNA sequencing was performed by the Life & Brain NGS Core Facility (Bonn, Germany) after QuantSeq 3'-mRNA Library Prep/ Lexogen enrichment at NovaSeq 6000 with a depth of 20 M reads. The raw data was processed using nf-core/rnaseq v3.12.0 (https://doi.org/10.5281/zenodo.1400710) of the nf-core collection of workflows[108]. The pipeline was executed with Nextflow v23.04.1[109]. The complete data set is available on the Gene Expression Omnibus repository (GEO accession number GSE297557). Differential expression analysis was performed using DESeq2 package[110] in R v4.4.3, applying an alpha of 0.05. Genes were considered differentially expressed at an adjusted $p$-value lower than 0.1. Heatmaps were generated from the DESeq data set after z-normalisation using the package pheatmap[111] and a custom-curated gene list. Functional enrichment of terms related to differentially expressed genes was performed using the g:Profiler online tool[112] using the terms GO molecular function, GO cellular component and GO biological process as well as the pathways KEGG, Reactome and WikiPathways. For comparison, publicly available data on the ageing human RPE was obtained from Butler et al.[53] and Cai et al.[52].

### Immunoblotting

Samples were harvested using RIPA buffer[113] with phosphatase inhibitors PhosSTOP™ (Roche, 4906845001) and were subjected to DNA digestion for 15 min by Pierce Universal Nuclease for cell lysis (Thermo Fischer, 88700) prior to addition of Lämmli buffer[114]. Samples were then heated to 95 °C for 7 min and ca. 10 µg were loaded onto a 10 to 14% (depending on target protein molecular weight) polyacrylamide gel[115] for SDS-PAGE together with a ProSieve Quadcolor Protein Marker (Lonza, 00193837). Proteins were blotted onto Immobilion-P PFDV membrane (Thermo Fischer 88518) and labelled with primary and secondary antibodies for different targets. The following primary antibodies were used: phospho-myosin light chain 2 (Ser19) (Cell Signalling Technology, Cat# 3671) at 1:1000, ezrin (Abcam, Cat# ab4069) at 1 µg/ml and phospho-ezrin (Thr567) (Invitrogen, Cat# PA5-37763) at 1 µg/ml. Secondary antibodies included: HRP anti-rabbit IgG (Abcam, Cat# ab99696) at 1:5000 and HRP anti-mouse IgG (H + L) (Jackson ImmunoResearch/Dianova, Cat# 115-035-003) at 1:5000. Bands were detected at the Fusion Solo (Vilber Lourmat) with the FusionCapt Advance Software after detection with the Pierce™ ECL Western blotting substrate (Thermo Fischer, 32106). The signal was quantified in Fiji[107] by using equal-sized region of interest (ROI) for all bands and measuring the mean grey value. The measurements for the proteins of interest were normalised to the housekeeping protein GAPDH.

## Modulation of actin nucleator and myosin II activity

Actin nucleation by Arp2/3 was modulated by using CK-666 (Tocris Bioscience, #3950) at 500 nM in medium. After 1 h incubation, the monolayers were characterised by nanoindentation. For phagocytosis studies, the monolayers were incubated for 30 min with 500 nM CK-666, followed by incubation with the photoreceptor fragments and the bridging molecules in CK-666 containing medium for 4 h. For modulating formin activity, 20 μM SMIFH2 (Tocris Bioscience, #4401) was used instead keeping the same timing and procedure. DMSO at a paired concentration was used as carrier control. For the incubation with different SMIFH2 concentrations, a series of dilutions was prepared starting from a 100 μM solution, resulting in a dilution series of 100, 40, 20 and 5 μM concentrations. For nanoindentation and POS functional studies, the same timing and procedure were used as described above with DMSO as a carrier control. Myosin II inhibition was performed using 5 μM blebbistatin (Abcam, ab120425) for 1 h before nanoindentation and for 30 min before POS addition in blebbistatin-containing medium (see above).

## Analysis of extrusion events

Extrusion events were quantified from live cell imaging of apoptosis induction on SiR-Actin (Spirochrome, SC001) labelled cells. The last 15 hours of apoptosis induction were used for the quantification which relies on extrusion events being visible as very bright spots. The video was aligned to correct for stage shifts, smoothed using a Gaussian filter and overlayed in a maximum intensity z-projection in Fiji. The resulting image was thresholded to generate a binary mask and extruded cells were counted using the particle analysis tool in Fiji.

## Analysis of cell density

Cell density was quantified from snaps of nuclear staining obtained on epi-fluorescence microscope at 20x magnification. Three to four snaps were used per condition per experiment. Nuclear signal was smoothed using a Gaussian filter, then detected by thresholding. The binary mask was subjected to watersheding and the number of nuclei was counted by using the particle analysis tool in Fiji. Density was calculated by dividing the obtained nuclei count to the area of the field of view. Data originates from at least three independent experiments.

## Analysis of cell height

Cell height was determined from super-resolution z-stacks of F-actin staining in cross-sectional view. Cells were imaged fixed after mounting in Mowiol, which leads to lower absolute height compared to the height detectable in live-cell imaging. Ten cells per stack were measured with the line tool in three stacks per condition from three independent experiments.

## Analysis of cell outline segmentation and shape factor calculation

Cellular outlines from F-actin labelling were segmented using Cellpose v.3.0.11[116]. The generated regions of interest (ROI) were imported in Fiji and the cell area and perimeter were measured to calculate the dimensionless shape factor[117,118].

## Analysis of stress fibres

Cells with stress fibres were manually counted using the multipoint tool in Fiji and normalised to the cell number as calculated after segmentation of cells via Cellpose v.3.0.11.

## Analysis of microvilli length

Microvilli length was characterised from cross-sectional views of ezrin labelling in Fiji. Briefly, at least three cross-sections from at least three stacks per condition from three independent experiments were blurred with a Gaussian blur and a threshold was applied to capture the structure. The resulting masks were selected via the magic wand tool

and an eclipse fit was performed. The minor axis was used as an approximate measure of the microvilli length.

## Photoreceptor fragment phagocytosis analysis

The photoreceptor fragment phagocytosis was characterised using the Software Imaris v10.2.0 (Oxford Instruments). Briefly, the actin labelling was used to create a surface capturing only the apical actin portion of the monolayer. The average z-position of the surface was determined as a cuff-off height to distinguish internal vs external particles. Separately, photoreceptor fragments were segmented as a surface and the cut-off height was used to distinguish between internalised and external fragments. The count and volume of the fragments of both populations was recorded and used for further analysis.

## Quantification of apical actin area

The apical actin area of $300 \times 300$ pixel field of view was segmented from z-stacks of phalloidin staining during photoreceptor internalisation as a surface in Imaris v10.2.0 (Oxford Instruments). For the region of interest, only a set number of slices including the same upper portion of the cell monolayer were used. The covering area, as internally determined by the software, was then noted.

## Quantification of vinculin junctional to cytoplasmic ratio

Junctional signal of vinculin was quantified as a ratio of junctional to cytoplasmic mean grey value in single optical sections. For this, paired F-actin signal was used to segment cell outlines via Cellpose v.3.0.11 as described above. The generated ROIs were imported in Fiji and a custom macro was used to slim-down the ROIs, invert the mask, watershed the signal and generate ROIs that encompass the junctional regions around cells. The generated junctional ROIs were used to measure grey value as an average per image. The slimmed-down cell ROIs were used to measure the cytoplasmic signal. The average values of junctional and cytoplasmic signals per image were used to generate the junctional to cytoplasmic vinculin measurements.

## Quantification of junctional pMLC2

Paired samples from the same experiment were stained with the same antibody solution and imaged with the same imaging settings. In single optical sections at the junctional level, pMLC2 signal was thresholded in Fiji making sure the junctional signal is captured. The mean grey value was measured only within the thresholded region. At least three images were measured per condition per experiment.

## Quantification of MRTFA

MRTFA was quantified as nuclear to cytoplasmic ratio of the mean grey value. For this, paired cell nuclei signal (DAPI staining) was segmented via Cellpose v.3.0.11 to generate nuclear ROIs. Nuclear MRTFA was measured as average of the mean grey value of all nuclei ROIs within the image. Cytoplasmic MRTFA was measured after setting nuclear ROIs pixels as zero and thresholding the rest of the image. Then, the mean grey value of the thresholded, non-zero region was measured.

## Brillouin microscopy

Brillouin imaging was performed using a commercial setup based on an inverted Nikon Ti-2 Eclipse microscope using a vertically polarised, continuous-wave laser (Cobolt Flamenco, $\lambda = 660$ nm, ~100 mW) beam with a temperature-tuned air-spaced etalon (LightMachinery) attenuated by 0.6 optical density (OD) filter, resulting in approximately 9 mW incident power at the sample. Light was coupled into a confocal circulator (LightMachinery, HF-13311) and directed to the microscope through a polarising beam splitter and a quarter wave plate (QWP) yielding circularly polarised light. Measurements were performed on a motorised translation stage (Prior Scientific, H117E2NN) using an air objective (Nikon S Plan Fluor 20X NA 0.45), achieving an estimated 0.73 μm lateral and 6.52 μm axial resolution. Brightfield and

fluorescence imaging were performed sequentially using a Teledyne sCMOS camera and epi-fluorescence illumination (Nikon D-LEDi).

Backscattered Rayleigh and Brillouin signals were collected with the same objective and routed back through the confocal circulator. The Brillouin signal was acquired using a LightMachinery HyperFine spectrometer (HF-8999-PK-660), equipped with a "pump killer" (PK) module with an air-spaced, pressure-tuned etalon to suppress the elastically scattered Rayleigh light and laser line reflection (~60 dB attenuation). Frequency-shifted Brillouin Stokes and anti-Stokes components were directed into the spectrometer, comprised of a virtually imaged phase array (VIPA) and a reflective echelle diffraction grating for spectral dispersion. The cross-dispersed orders were focused onto the CMOS camera sensor (Hamamatsu ORCA-Fusion, C14440-20UP) simultaneously acquiring the full spectrum. The typical acquisition time of a single spectrum is 500 ms, with a spectral resolution <1 pm.

Spectral data acquisition and raster scanning was facilitated with LightMachinery LabView-based software. The x-y pixel coordinates on the CMOS sensor are converted to wavelength (or frequency) and a Lorentzian fit is applied over each Brillouin peak as described by equation (1).

$$I(\omega) = \frac{I_0}{\pi} \frac{\Gamma \omega_B^2}{(\omega^2 - \omega_B^2)^2 + \Gamma^2 \omega_B^2} \tag{1}$$

The final Brillouin shift $\omega_B$ and linewidth $\Gamma$ values for a given spectrum were computed by averaging the corresponding values from the Stokes and anti-Stokes peaks forming a single voxel in the raster scan. Z-scans were acquired as a $3 \times 3$ matrix in a step size of 20 μm and z-step of 2 μm, z-range of 80 μm, while apical and basal scans were acquired over the region of $50 \times 50$ μm with a lateral step size of 1.5 μm. The z-position of apical and basal Brillouin scans within the monolayer was selected based on the appearance of the cells in brightfield and SiR-actin fluorescence images. The scans were limited to 15 minutes to obtain an optimal raster scan resolution while maintaining the cell viability. It should be noted that although the average laser power at sample is within the range used in biological samples, by comparing brightfield images taken before and after the Brillouin scan, some deformation of the sample was observed. This could have occurred due to the heat of the laser affecting the gel or the cells as the RPE cells' pigmentation contributes to the linear laser light absorption.

### Brillouin shift data analysis

For segmentation, the Brillouin shift maps from basal layers were linearly scaled and converted into 8-bit grayscale images, proportionally reflecting the original Brillouin shift values. The grayscale images were segmented using Cellpose v.3.0.11 to identify the individual structures within the images. Following segmentation, two populations were defined: "core", corresponding to segmented regions, and "interstitium", corresponding to areas between them. These modified images were then imported into a custom-written MATLAB script, where appropriate binary masks were created and applied to the original Brillouin shift maps to separate the "core" and the "interstitium". For each raster scan, the mean Brillouin shift and the standard deviation were calculated separately for the two regions. The $3 \times 3$ raster z-scans resulted in nine representative z-axis line profiles per region, capturing mechanical transition across the sample. Each profile typically displayed a stiffness change from the hydrogel to the monolayer and then from the monolayer to the surrounding cell media. For each sample, 5 of these scans were obtained.

The Brillouin Shift data were imported into a custom-written MATLAB script to identify the local peaks or, where absent, plateaus in the Brillouin shift profiles. Each line profile was smoothed with a Savitzky-Golay filter, the peak was detected using MATLAB's findpeaks function, with a minimum width imposed to prevent false peak

identification. A gradient of the filtered z-line was taken. To refine peak localisation, a Gaussian fit was applied across a range of filtered Brillouin shift values. The fitting range was bound by the z-positions at which the local maximum and the minimum of the derivative of the z-line was within an appropriate range from the previously identified peak. The fitted peak value and position were retrieved. In the case of no peak detection, the code identifies the onset of a drop in the plateau by considering the change in the derivative over a moving average window. The corresponding plateau value was then recorded. The performance of the script was checked for individual z-profiles to exclude outliers.

### Statistics and reproducibility

Statistical analysis was performed using GraphPad Prism 10 on paired averaged values originating from at least three independent biological experiments. Datapoints from at least three technical replicates or from matrix measurements within the samples, epithelial monolayers, were averaged into representative values from a biological replicate. Assuming normal distribution, the data was tested with a paired, two-sided test depending on experimental design with statistical tests and number of biological replicates for each graph given in the captions of the figures. Samples were randomly assigned to experimental conditions. Data were excluded only if obvious sample damage occurred or if technical failures prevented reliable analysis. No other data were excluded. The investigators were not blinded to allocation during experiments or outcome assessment.

### Reporting summary

Further information on research design is available in the Nature Portfolio Reporting Summary linked to this article.

## Data availability

All data are available in the main text, supplementary materials or in the Source Data file. Furthermore, the complete bulk RNA-sequencing dataset is available in the Gene Expression Omnibus repository (GEO) under accession number GSE297557. Source data are provided with this paper.

## Code availability

The custom code for identifying local peaks in Brillouin shift z-profiles is available at the following link: https://doi.org/10.5281/zenodo.18888268. The code for RNA sequencing raw data processing (nf-core/rnaseq v3.12.0) is available under the following link: https://doi.org/10.5281/zenodo.1400710.

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

## Acknowledgements

This work in Di Russo's laboratory was supported by a grant from the Interdisciplinary Centre for Clinical Research within the Faculty of Medicine at the RWTH Aachen University and a grant from the German Research Foundation (DFG; RU 2366/3-1). Furthermore, J.D.R. and W.W. are supported by the DFG within 363055819/GRK2415. The collaboration with Vassalli's lab for Brillouin microscopy was supported by an RS Mac-Donald Facility Access funding grant by the Scottish Universities Life Sciences Alliance (SULSA), a travel grant by the German Academic Exchange Service (DAAD) programme number 57745342 and a travel grant by Boehringer Ingelheim Fonds. T.P. and A.N.K. are supported by Add-on Fellowships of the Joachim Herz Foundation. We would like to acknowledge all the members of the REMeD group in the past years for their contributions to the discussion of the work. Moreover, we appreciated Adam Breitscheidel's technical support in graphic design, Gloria's excellent help in imaging and insightful feedback from Rudolf Leube. We would like to acknowledge the support of Chris Davis from the Centre for Virus Research of the University of Glasgow for providing a lab space in Garscube campus to enable AAV work needed for the sample preparation for Brillouin microscopy. This work was supported by the Two-Photon Imaging Facility of the Interdisciplinary Center for Clinical Research (IZKF) within the Faculty of Medicine at RWTH Aachen University.

## Author contributions

J.D.R. conceived and supervised the project. T.P. designed and performed all the experiments. G.A., A.N.K. and T.P. performed Brillouin measurements. G.A. and A.N.K. performed Brillouin analysis. M.E.M. performed initial transcriptomics data analysis and advised on the transcriptomic exploration. RR supported the immunoblotting experiments. SS supported with confocal imaging. S.L.S.Y. supported the establishment of viral transduction. W.W. and M.V. supported the work with expertise and advice. J.D.R. and T.P. wrote the manuscript, and all authors discussed and commented on it.

## Funding

## Competing interests

The authors declare no competing interests.
