## [Transparent Peer Review file · Nature Communications]

Cell loss disrupts mechanical homeostasis to drive retinal pigment epithelium ageing-like phenotype in vitro

Corresponding Author: Dr Jacopo Di Russo

Version 0:

Reviewer comments:

Reviewer #1

(Remarks to the Author)

Reviewer report on "Tissue organization alone drives age-like changes in a postmitotic epithelium" by Piskova et al. (2025)

Tissue mechanics (including elasticity, viscosity, pressure, compliance, among others) play a critical role in cancer development, influencing how tissue grows, functions, and responds to external mechanical cues. This study explores how cellular organization alone—without genetic mutations or external stressors—can mimic ageing effects in the retinal pigment epithelium (RPE), a crucial postmitotic tissue in the eye. The authors used an in vitro model system based on human induced pluripotent stem cell-derived RPE (iRPE) cultured on soft hydrogels. The authors showed that the described AgeD-iRPE model in vitro model recapitulates structural ageing features previously shown in human RPE tissues including (1) larger cell area and reduced cell height, (b) shortened and disorganizes microvilli, and (c) transcellular actin stress fibers. Then, the authors proceeded to investigate if the model recapitulates impaired internalization of outer segment fragments, and indeed they showed that AgeD-iRPE monolayers exhibit deficient internalization. Then, the authors hypothesized that internalization deficiency could be due to active remodeling of the actomyosin network. The authors performed RNA sequencing to investigate changes in mRNA expression of actin regulatory proteins and observed a vast amount of upregulated and downregulated actin associated genes. Next, the authors investigate changes in mechanical properties by nanoindentation and Brillouin microscopy. The authors observed a significant increase in elasticity by Young's modulus and storage modulus that seem to correlate with an apparent increase in phosphorylated myosin (pMLC2) and vinculin expression. Surprisingly, with Brillouin microscopy the authors observed an opposite mechanical behavior with decrease Brillouin shift indicating softening. Last, the authors performed small molecule pharmacological perturbations to inhibit actin-based nucleation either inhibition of branched actin nucleation (ARP2/3) by CK-666 or inhibition of linear actin nucleation (formins) by SMIFH2. The authors observed that by modulating actin nucleators phagocytosis is impacted (a) Arp2/3 inhibition (via CK-666) improved internalization in AgeD-RPE whereas (b) formins inhibition (via SMIFH2) increase phagosome size in control iRPE.

Altogether, this manuscript seems to describe a new in vitro model to investigate RPE ageing, however there are technical concerns in some conducted experiments and some controls are missing. The current manuscript version is rather superficial and of incremental nature. I believe a revised version of this manuscript will be more suitable to be submitted for publication consideration in a specialized cell biology journal. Therefore, I do not recommend the publication of this manuscript in Nature Communications.

Comments:

1. Figure 1: The transcellular stress fibers the authors describe in the manuscript are not clearly visible. Even trying to distinguish the transcellular stress fibers of zoomed-in images from the AgeD-iRPE that has been highlighted with arrows are hard to see (Fig. 1F).
2. Figure 1: In addition, to this reviewer looking at the F-actin images from both control iRPE and AgeD-iRPE it seems that there is the same amount of transcellular stress fibers on both conditions (Fig. 1E). Looking at these images alone is hard to compare both conditions and judge if there is indeed an increase of 10 folds more percentage of transcellular stress fibers. I suggest the authors add zoomed-in images from the control iRPE in this figure (Fig. 1F) next to the zoomed-in images of Aged-iRPE so readers can compare both conditions and judge the changes by themselves.

3. Figure 4: Can the authors provide more detailed explanations of why the Brillouin shift is significantly lower in AgeD-iRPE when compared to control iRPE and how this is contradictory to the obtained microindentations results. There is published work correlating changes in microindentation based mechanical properties changes with Brillouin shift and have shown that when it is observed an increase in stiffness measured by microindentation measurements this corresponds to a significant increase in measured Brillouin shift (Weber et al. *Physical Biol.* 14, 2017; Scarcelli et al. *Nat. Methods* 12, 2015). Therefore, the results shown in this work are contradictory with published work and need to be clearly clarified and discussed.

4. As briefly discussed in the discussion section of this manuscript, SMIFH2 has been shown to have nonspecific off target effects and on many occasions strongly inhibits non muscle myosin II myosin ATPase activity (Nishimura et al. *J. Cell Sci.* 134, 0221). In addition, it seems to inhibit several other non-muscle myosin types. Therefore, without a targeted perturbation of formins protein expression to validate the SMIFH2 results or by performing SMIFH2 experiments at different concentrations it is hard to judge if these results are valid or artefactual.

5. Continuing on this line about the formin inhibition results, a lot of published work has shown an opposite effect to what the author describes in this manuscript. Previous work (Mittal et al. *Commun. Mater.* 5, 2024; Efremov et al. *BBA-Mol. Cell Res.* 1853, 2015) inhibiting formins in many different cell types including (3T3 fibroblasts, DU145 cells, and Vero cells) has shown a drastic softening effect. Therefore, can the authors comment on this surprising contradiction.

6. Control experiments are missing. The authors must perform experiments on AgeD-iRPE treated with SMIFH2 and control iRPE treated with CK-666 to contrast the effects of inhibition in both conditions.

7. Minor comment. On several occasions throughout the paper the authors make statements such as "first-of-a-kind", "for the first time", and others. I suggest the authors tone down the manuscript and avoid using such statements.

Reviewer #2

(Remarks to the Author)

In this study, Piskova and colleagues developed and characterized an in vitro aging model of retinal pigmented epithelium (RPE) to investigate how actin contributes to the reduced phagocytosis of photoreceptor outer segments during aging. This work presents novel mechanisms that will be of interest to a broad audience.

The experimental design is sound with appropriate controls, and the methodological approaches are well-suited to address the research questions. The data analyses appear rigorous, and the conclusions are supported by the presented evidence. However, the manuscript has several presentation issues that limit its accessibility. The abstract lacks clarity due to ill-defined terminology, and the results section focuses heavily on data presentation but provides insufficient interpretation of the biological significance of the observations.

Overall, I support this manuscript for publication, but the authors must address the clarity and presentation issues outlined below to make their important findings accessible to the broad readership of Nature Communications.

Major Comments

1) The novelty and impact of this study are difficult to understand from the abstract, primarily due to the use of numerous ill-defined and unconventional terms. For example, what do the authors mean by "mechanical homeostasis" and "structural ageing"? Similarly, the term "impaired-plasticity" requires clarification. More critically, the main research question addressed in this study is unclear. The authors state they are investigating "the connection between age-related structural changes and mechanical homeostasis," but it is not evident how this relates to their proposed answer: "a mechanistic link between cytoskeletal adaptation and phagocytic dysfunction in ageing retinal pigment epithelium." I recommend rephrasing the abstract to clearly define key terms, explicitly state the research question, and demonstrate how the findings address that question.

2) The novelty of the findings is not sufficiently emphasized throughout the manuscript. This may be because much of the data focuses on characterizing the AgeD-iRPE model, showing results consistent with in vivo data such as reduced cell height, shortened microvilli, actin cytoskeletal alterations, and decreased phagocytic ability. While the cellular and subcellular mechanical characterization of aging RPE is novel, the authors should better highlight the importance of these mechanical measurements and their meaning.

A few key findings can be emphasized more clearly: first, the establishment of an in vitro aging RPE model, which represents a significant methodological advance for the field. Second, the challenge to the current notion of cellular 'hypertrophy' is particularly noteworthy. The results demonstrate that aging involves not cell volume enlargement but rather 'tissue stretching from within,' pointing to individual cell volume conservation. This novel concept, which challenges traditional views of cellular aging, should be highlighted more prominently in the results section rather than being buried in the discussion.

3) Related to point #1, the authors frequently use poorly defined terminology that obscures their interesting findings. Several examples require clarification:

- What do the authors mean by "a new mechanical equilibrium and homeostasis" and "actin-myosin network plasticity" (line 368)?
- How should readers interpret "a shift in the range of mechanical equilibrium" (line 431)?
- What is meant by "intercellular reinforcement" (line 481)?

4) The results section is difficult to follow because while data are presented and quantifications are explained, there is insufficient interpretation of what the results mean and why these observations are important. The authors need to better explain the significance of their findings. Several examples illustrate this issue:

- What do the nanoindentation data mean (lines 242-252)?
- Why does the difference in Brillouin shift at different sections (apical to basal) lead to the conclusion that "the apical sides of the cells undergo the most pronounced reorganisation" (lines 293-306)? Moreover, what specific type of 'reorganisation' are the authors referring to, and why is this important?
- What is the significance of the segmented Brillouin shift data (lines 311-317)? Why should readers care about "the presence of loosely interdigitated basal membranes"?
- Why is the observation that "age-associated RPE phenotype has a different biomechanical state of the tissue" important?
- What does the finding that "bulging cells showed a more irregular apical morphology (Fig. 5D)" imply for RPE function?

Why do the authors highlight this observation? A similar question applies to the statement that "the treated AgeD-iRPE showed some degree of 'bulging' during phagocytosis, which was more noticeable compared to the untreated AgeD-iRPE (Fig. 5H)."

Minor Comments

5) Fig. 6: Why does the middle panel showing top views highlight the observed characteristics only in the central cell and not in the neighboring cells as well?

6) Supp. Fig. 3A(2): The screenshot from an application is not appropriate for a scientific publication.

7) The statement that "a balance between basal cell-ECM adhesion and apical cell-cell adhesion exists and is normally well conserved (78–80)" (line 456) requires more detailed discussion.

Reviewer #3

(Remarks to the Author)

This is a well executed study on the physiological changes of the retinal pigmented epithelial monolayer in response to 2% apoptosis in vitro. The quality of the experimental design, the chosen assays and the execution is all excellent. The framing of the experiments is where this reviewer had much issue. There are flaws in the data interpretation and conclusions which prohibit publication or require revision. They make bodacious claims as to what their results signify. To so without qualifications of which there are many. They also use duplicitous naming of their model attempting to more closely align what they are studying (epithelial monolayer response to apoptosis) to what they claim to be studying (epithelial monolayer aging). Moreover there is no qualifications with regards to the differences between in vivo and in vitro conditions nor the known differences between iPSC-RPE compared to native RPE.

The experiments detailed in this manuscript does not support the title. "Tissue alone" is too vague for any understanding "alone" claim has not been supported by the evidence of this MS. The in vitro model is lacking many of the conditions in a live aging organism and therefore the conclusions can only be made in that model. Therefore, the title should qualify the statement. "within an in vitro model..."

The abstract is too general and does not discuss what is done, what results were found and what conclusions from those results can be made.

The in vitro aging may exhibit similar morphological changes for very different reasons. Lack of vascular support, lack of the ability to maintain ionic balance of the media apically and basolaterally, no differential in nutrient access apically to basolaterally. These over time can also lead to a decrease in physiological function and have nothing to do with the native environment.

Hydrogels do not have a composition resembling native RPE environment. Water content, diffusion gradient,

The induced apoptosis is too synthetic to model native RPE aging. Therefore what is being modeled is the cellular response to apoptosis.

There is no age dependent condition and therefore, the way the experiments are described is a misnomer.

Replace AgeD RPE with apoptosis responsive ARD-RPE and then the manuscript is on a more accurate foundation from which to draw conclusions. To mix apoptosis with age is duplicitous.

The lack of proliferation in response to apoptosis is an interesting response that warrants further study. What is missing is a spectrum of apoptosis responses from 0 to 50% at regular intervals. Is there an apoptosis % that proliferation can be triggered? If so, what are the mechanisms governing this? Is there another more natural apoptosis approach that can support these findings or must it be the manipulation of caspase to lead to these changes within this timeframe?

The results are informative so long as the authors are more honest with their claims.

There is sufficient detail in the methods section to reproduce the experiments.

Commend them on the proper protocol for assaying phagocytosis.

In general experimental execution and data collection quality is high.

Version 1:

Reviewer comments:

Reviewer #2

(Remarks to the Author)

In this revised manuscript, the authors have addressed most of the points I raised in my original review. I appreciate the authors' thorough responses.

One minor comment relates to my original point #4: Why should readers care about 'the presence of loosely interdigitated basal membranes'? The authors responded that 'We thought mentioning the loosely interdigitated basal membrane may provide a likely explanation for any questions readers may have regarding the data.' However, without context, readers (including myself) remain confused about what this observation means. To clarify this point, I suggest providing at least one specific example of a question that readers might have about your data that would be answered by this basal membrane observation and explain how this structural feature addresses that question in the text. This will help readers understand the relevance of this finding to your overall results. Since this observation is featured in both the text and figures, readers need a clear explanation of its relevance.

Once this issue is addressed, I support this manuscript for publication.

(Remarks on code availability)

Reviewer #3

(Remarks to the Author)

The authors satisfied all of the major issues in my previous review. I support acceptance.

(Remarks on code availability)

Point-by-point response

Reviewer #1 (Remarks to the Author)

Reviewer report on “Tissue organization alone drives age-like changes in a postmitotic epithelium” by Piskova et al. (2025)

Tissue mechanics (including elasticity, viscosity, pressure, compliance, among others) play a critical role in cancer development, influencing how tissue grows, functions, and responds to external mechanical cues. This study explores how cellular organization alone—without genetic mutations or external stressors—can mimic ageing effects in the retinal pigment epithelium (RPE), a crucial postmitotic tissue in the eye. The authors used an in vitro model system based on human induced pluripotent stem cell-derived RPE (iRPE) cultured on soft hydrogels. The authors showed that the described AgeD-iRPE model in vitro model recapitulates structural ageing features previously shown in human RPE tissues including (1) larger cell area and reduced cell height, (b) shortened and disorganizes microvilli, and (c) transcellular actin stress fibers. Then, the authors proceeded to investigate if the model recapitulates impaired internalization of outer segment fragments, and indeed they showed that AgeD-iRPE monolayers exhibit deficient internalization. Then, the authors hypothesized that internalization deficiency could be due to active remodeling of the actomyosin network. The authors performed RNA sequencing to investigate changes in mRNA expression of actin regulatory proteins and observed a vast amount of upregulated and downregulated actin associated genes. Next, the authors investigate changes in mechanical properties by nanoindentation and Brillouin microscopy. The authors observed a significant increase in elasticity by Young's modulus and storage modulus that seem to correlate with an apparent increase in phosphorylated myosin (pMLC2) and vinculin expression. Surprisingly, with Brillouin microscopy the authors observed an opposite mechanical behavior with decrease Brillouin shift indicating softening. Last, the authors performed small molecule pharmacological perturbations to inhibit actin-based nucleation either inhibition of branched actin nucleation (ARP2/3) by CK-666 or inhibition of linear actin nucleation (formins) by SMIFH2. The authors observed that by modulating actin nucleators phagocytosis is impacted (a) Arp2/3 inhibition (via CK-666) improved internalization in AgeD-RPE whereas (b) formins inhibition (via SMIFH2) increase phagosome size in control iRPE.

Altogether, this manuscript seems to describe a new in vitro model to investigate RPE ageing, however there are technical concerns in some conducted experiments and some controls are missing. The current manuscript version is rather superficial and of incremental nature. I believe a revised version of this manuscript will be more suitable to be submitted for publication consideration in a specialized cell biology journal. Therefore, I do not recommend the publication of this manuscript in Nature Communications.

We thank the reviewer for the constructive feedback. We respectfully disagree that the work is incremental, as the manuscript identifies a previously unrecognised mechanism of homeostasis in the RPE and more broadly in postmitotic tissues, and demonstrates its consequences for ageing-related dysfunction.

We have carefully addressed all specific comments and revised the manuscript to clarify the conceptual advances, strengthen the methodological description, and improve the integration of findings across disciplines. We believe these revisions now more clearly articulate the impact of our work within mechanobiology, cell biology, and retinal physiology.

Comments:

1. Figure 1: The transcellular stress fibers the authors describe in the manuscript are not clearly visible. Even trying to distinguish the transcellular stress fibers of zoomed-in images from the AgeD-iRPE that has been highlighted with arrows are hard to see (Fig. 1F).

Figure 1f was improved by enlarging the images and adding a zoom-in of a single cell and stress fiber to better highlight the phenomenon.

2. Figure 1: In addition, to this reviewer looking at the F-actin images from both control iRPE and AgeD-iRPE it seems that there is the same amount of transcellular stress fibers on both conditions (Fig. 1E). Looking at these images alone is hard to compare both conditions and judge if there is indeed an increase of 10 folds more percentage of transcellular stress fibers. I suggest the authors add zoomed-in images from the control iRPE in this figure (Fig. 1F) next to the zoomed-in images of Aged-iRPE so readers can compare both conditions and judge the changes by themselves.

Figure 1f now displays both conditions side by side, which indeed allows for better comparison.

3. Figure 4: Can the authors provide more detailed explanations of why the Brillouin shift is significantly lower in AgeD-iRPE when compared to control iRPE and how this is contradictory to the obtained microindentations results. There is published work correlating changes in microindentation based mechanical properties changes with Brillouin shift and have shown that when it is observed an increase in stiffness measured by microindentation measurements this corresponds to a significant increase in measured Brillouin shift (Weber et al. Physical Biol. 14, 2017; Scarcelli et al. Nat. Methods 12, 2015). Therefore, the results shown in this work are contradictory with published work and need to be clearly clarified and discussed.

We understand the reviewer's concern, and we are aware of the apparent contradiction, but believe the results are reconcilable because of two major factors:

1. Indentation-based and optical elastography-based methods differ fundamentally in how they assess mechanical properties.

Specifically, they differ in probing frequency, invasiveness (mechanical vs. optical), and hydration influence. Notably, a recent study demonstrated that hydration levels in heterogeneous materials, such as cells, greatly affect the longitudinal elastic modulus measured by Brillouin microscopy. However, indentation-based approaches like AFM and nanoindentation are not impacted by hydration because

they allow water to expand transversely during measurement (ref: 10.1021/acsphotonics.5c00808).

2. In the nanoindentation vs Brillouin microscopy, we probe the tissue at different length scales.

During nanoindentation, we use a 20 μm in diameter sphere that indents into a few cells at a time, often impacting on cell-cell contacts. As we were interested in probing tissue-level mechanics, so important for epithelia, we did not segregate between measurements hitting purely on a cell vs measurements hitting on cell-cell junctions. Thus, our nanoindentation results heavily reflect the stiffness of the cell-cell junctions, counteracting the deformation. On the contrary, as our Brillouin microscopy measurements relied on point-by-point laser scanning within the apical and basal optical planes of the monolayers, the cell cortices may have had a higher contribution to the obtained average values rather than the cell-cell contacts, as the cortical, non-junctional portion of the images occupies more pixels.

Lastly, direct comparisons between new methods like Brillouin microscopy and established ones such as nanoindentation are important to reveal discrepancies and encourage careful interpretation, since these techniques rely on fundamentally different principles. As also recently highlighted in a work on retinal tissue (DOI: 10.1088/2515-7647/ad5ae3), as complementary, both techniques are necessary for a comprehensive characterisation of a system.

To avoid diverting attention from the main message of our work, we did not emphasise the apparent contradiction with other biological systems (e.g., Scarcelli et al. Nat. Methods 12, 2015) or length scales (Weber et al. Physical Biol. 14, 2017) when comparing Brillouin microscopy and nanoindentation. Nonetheless, we explicitly mentioned that a direct comparison cannot be made (lines 429-430) and that any difference may be attributed to changes in microvilli density and apical actin remodelling (lines 430-432 and 254-255).

4. As briefly discussed in the discussion section of this manuscript, SMIFH2 has been shown to have nonspecific off target effects and on many occasions strongly inhibits non muscle myosin II myosin ATPase activity (Nishimura et al. J. Cell Sci. 134, 0221). In addition, it seems to inhibit several other non-muscle myosin types. Therefore, without a targeted perturbation of formins protein expression to validate the SMIFH2 results or by performing SMIFH2 experiments at different concentrations it is hard to judge if these results are valid or artefactual.

We thank the reviewer for the valuable comment and agree that additional perturbation experiments were necessary. We have now included treatments with SMIFH2 at multiple concentrations, as well as the Myosin II inhibitor blebbistatin as a positive control. The data are now shown in Suppl. Fig. 8j–u and presented in lines 314-338 and discussed in lines 491-493.

Interestingly, the effects quantified following SMIFH2 treatment were not consistent with those observed using blebbistatin, particularly at 20 μM SMIFH2, which was the initial concentration used. Notably, SMIFH2 treatment increased iRPE stiffness only at 20 μM , which rules out a possible inhibitory effect on Myosin function, an effect that would

instead be expected to soften the monolayer, as demonstrated by blebbistatin and reported in the literature (e.g., Figure 2 in DOI: 10.3389/fcell.2022.946190).

5. Continuing on this line about the formin inhibition results, a lot of published work has shown an opposite effect to what the author describes in this manuscript. Previous work (Mittal et al. Commun. Mater. 5, 2024; Efremov et al. BBA-Mol. Cell Res. 1853, 2015) inhibiting formins in many different cell types including (3T3 fibroblasts, DU145 cells, and Vero cells) has shown a drastic softening effect. Therefore, can the authors comment on this surprising contradiction.

We thank the reviewer for raising this important point. The studies mentioned (e.g. Mittal et al. 2024; Efremov et al. 2015) primarily examined single cells. Our work focuses on postmitotic, static epithelium where cell-cell adhesions are in mechanical equilibrium. To our knowledge, the mechanical response to formin inhibition has not been previously characterised in this context. In our system, we believe that the increased stiffness reflects a redistribution of tension within mechanically integrated monolayers, rather than a cell-autonomous softening response.

6. Control experiments are missing. The authors must perform experiments on AgeD-iRPE treated with SMIFH2 and control iRPE treated with CK-666 to contrast the effects of inhibition in both conditions.

We have now included the missing controls after performing a new series of experiments, and the results are presented in the Sup. Fig. 8 c, d, g and h and in Sup. Fig. 9 c, d, g and h. In the text, we have described the results in the last chapter, starting from line 297.

7. Minor comment. On several occasions throughout the paper the authors make statements such as “first-of-a-kind”, “for the first time”, and others. I suggest the authors tone down the manuscript and avoid using such statements.

We have now changed our wording and toned down the narrative.

Reviewer #2 (Remarks to the Author):

In this study, Piskova and colleagues developed and characterized an in vitro aging model of retinal pigmented epithelium (RPE) to investigate how actin contributes to the reduced phagocytosis of photoreceptor outer segments during aging. This work presents novel mechanisms that will be of interest to a broad audience. The experimental design is sound with appropriate controls, and the methodological approaches are well-suited to address the research questions. The data analyses appear rigorous, and the conclusions are supported by the presented evidence.

However, the manuscript has several presentation issues that limit its accessibility. The abstract lacks clarity due to ill-defined terminology, and the results section focuses heavily on data presentation but provides insufficient interpretation of the biological significance of the observations. Overall, I support this manuscript for publication, but the authors must address the clarity and presentation issues outlined below to make their important findings accessible to the broad readership of Nature Communications.

We thank the reviewer for recognising the value of our work and conclusions. We agree that the text greatly benefited from the substantial revisions to clarify the significance of our findings. As detailed below, we have now made the suggested changes to the text.

Major Comments

1) *The novelty and impact of this study are difficult to understand from the abstract, primarily due to the use of numerous ill-defined and unconventional terms. For example, what do the authors mean by "mechanical homeostasis" and "structural ageing"? Similarly, the term "impaired-plasticity" requires clarification. More critically, the main research question addressed in this study is unclear. The authors state they are investigating "the connection between age-related structural changes and mechanical homeostasis," but it is not evident how this relates to their proposed answer: "a mechanistic link between cytoskeletal adaptation and phagocytic dysfunction in ageing retinal pigment epithelium." I recommend rephrasing the abstract to clearly define key terms, explicitly state the research question, and demonstrate how the findings address that question.*

We completely rewrote the abstract, clarifying the reviewer's points and avoiding niche terms.

- Now the term “*Mechanical homeostasis*” is understandable from the context of the first three sentences.
- The terms *plasticity* and *structural ageing* were removed or rephrased.
- The abstract contains the main question:
[... *Yet, how these adaptations relate to epithelial mechanical homeostasis and age-associated functional decline remains poorly understood...*]
- How the findings address the question:
[...*To establish the relationship between structural changes, mechanical homeostasis and function, we developed an in vitro reductionistic model mimicking age-related reduction in RPE cell density. Inducing large-scale apoptosis in postmitotic stem cell-derived RPE monolayers recapitulates structural hallmarks of aged tissue, such as reduced cell height, shortened microvilli and cytoskeletal reorganisation. This new structure acquires a new mechanical equilibrium, evidenced by tissue stiffening and enhanced junctional contractility. Functionally, the monolayers display impaired vision-supporting phagocytosis of photoreceptor outer segments. Mechanistically, modulation of actin nucleators, Arp2/3 and formins, demonstrates that apicolateral monolayer deformation is critical for phagocytosis and may be compromised in aged RPE...*]

2) *The novelty of the findings is not sufficiently emphasized throughout the manuscript. This may be because much of the data focuses on characterizing the AgeD-iRPE model, showing results consistent with in vivo data such as reduced cell height, shortened microvilli, actin cytoskeletal alterations, and decreased phagocytic ability. While the cellular and subcellular mechanical characterization of aging RPE is novel, the authors should better highlight the importance of these mechanical measurements and their meaning.*

A few key findings can be emphasized more clearly: first, the establishment of an in vitro aging RPE model, which represents a significant methodological advance for the field. Second, the challenge to the current notion of cellular 'hypertrophy' is particularly noteworthy. The results demonstrate that aging involves not cell volume enlargement but rather 'tissue stretching from within,' pointing to individual cell volume conservation. This novel concept, which challenges traditional views of cellular aging, should be highlighted more prominently in the results section rather than being buried in the discussion.

We appreciate the reviewer's feedback, which helped us sharpen the focus of our narrative. We have revised the manuscript accordingly to address the points raised.

- On the importance of the mechanical measurements:
We have added explanations in the introduction, lines 31-36, results, lines 271-281 and discussion, lines 453-468.
- On the in vitro model:
Although we align with the reviewer's enthusiasm to define this model as a possible ageing model, we want to be careful to consider this reductionistic *in vitro* system representative of the complex *in vivo* ageing process. We aimed to dissect only one aspect of ageing RPE, focusing more on the structural and mechanical perspectives. As correctly pointed out by reviewer #3, with our model, we study the direct response to cell-density reduction, which helps us to speculate on RPE ageing in the native environment. More studies are necessary to understand how far this approach can be considered as an ageing model. We included more clarification about our model and how it helps to make age-related conclusions in the introduction, lines 36-40, results, lines 124-126, and abstract: [*... To establish the relationship between structural changes, mechanical homeostasis and function, we developed an in vitro reductionistic model mimicking age-related reduction in RPE cell density...*]
- On the cellular hypertrophy:
On the same line regarding the validity of the *ageing model*, we are careful to conclude that the observations are directly transferable to RPE ageing. In fact, in the discussion, we want to explain what is happening in our model, which will be a stretch to compare directly with ageing RPE. However, we added a line in the discussion (lines 399 - 402) highlighting the significance for postmitotic epithelia homeostasis and emphasised the findings in the results (lines 123-124).

3) Related to point #1, the authors frequently use poorly defined terminology that obscures their interesting findings. Several examples require clarification:

- *What do the authors mean by "a new mechanical equilibrium and homeostasis" and "actin-myosin network plasticity" (line 368)?*
- *How should readers interpret "a shift in the range of mechanical equilibrium" (line 431)?*
- *What is meant by "intercellular reinforcement" (line 481)?*

We thank again the reviewer for their valuable inputs regarding the text. We have addressed the raised points in the revised manuscript version as follows:

- Regarding “mechanical equilibrium and homeostasis” and “actin-myosin network plasticity”:
We clarify the conclusions with an expanded paragraph in lines 359-367
- Regarding “a shift in the range of mechanical equilibrium”:
In the revised version, we introduce the concept of mechanical feedback loops and the interconnection with mechanical equilibrium and homeostasis early on in the introduction (lines 8-14), we highlight the change of mechanical state at the end of the corresponding result section (line 271-281), and we explain its implications for phagocytosis in the discussion (lines 494-497).
- Regarding “intercellular reinforcement”:
The sentence has been optimised for clarity, specifying that reinforcement is associated with an increase in contractility at the cell-cell junction (lines: 475 - 477).

4) The results section is difficult to follow because while data are presented and quantifications are explained, there is insufficient interpretation of what the results mean and why these observations are important.

We thank the reviewer for this helpful feedback on the clarity of our exposition of the work. Now we have added several more explanations throughout the results section.

The authors need to better explain the significance of their findings. Several examples illustrate this issue:

- What do the nanoindentation data mean (lines 242-252)?

We added a short explanation in lines 219-221.

- Why does the difference in Brillouin shift at different sections (apical to basal) lead to the conclusion that "the apical sides of the cells undergo the most pronounced reorganisation" (lines 293-306)?

Moreover, what specific type of 'reorganisation' are the authors referring to, and why is this important?

We expanded the data explanation at the end of the results section in lines 254-258.

- What is the significance of the segmented Brillouin shift data (lines 311-317)? Why should readers care about "the presence of loosely interdigitated basal membranes"?

We improved the clarification of why we did the segmentation of the interstitium vs the core, and the unexpected lower shift measured at the interstitium in lines 259-261 and 267-270.

We thought mentioning the loosely interdigitated basal membrane may provide a likely explanation for any questions readers may have regarding the data.

- Why is the observation that "age-associated RPE phenotype has a different biomechanical state of the tissue" important?

We expanded the explanation in a few points of the manuscript. In the results in lines 277-281 and lines 359-367, and in the discussion in lines 458-465.

- What does the finding that "bulging cells showed a more irregular apical morphology (Fig. 5D)" imply for RPE function? Why do the authors highlight this observation? A similar question applies to the statement that "the treated AgeD-iRPE showed some degree of 'bulging' during phagocytosis, which was more noticeable compared to the untreated AgeD-iRPE (Fig. 5H)."

The rationale for investigating formins vs. Arp2/3 function in iRPE and dr-iRPE stems from the apical F-actin topography observed during phagocytosis (Fig. 2h). We explain this in lines 287-295. With the expanded explanations in the results, lines 359-367, and in the discussion, lines 458-465, we believe this is now clearer.

Minor Comments

5) Fig. 6: Why does the middle panel showing top views highlight the observed characteristics only in the central cell and not in the neighboring cells as well?

We thank the reviewer for the observation and have now improved Figure 6 accordingly.

6) Supp. Fig. 3A(2): The screenshot from an application is not appropriate for a scientific publication.

We have now substituted the low-resolution exported panel with a better-resolved graph.

7) The statement that "a balance between basal cell-ECM adhesion and apical cell-cell adhesion exists and is normally well conserved (78–80)" (line 456) requires more detailed discussion.

We added some more detailed discussion in lines 441-446.

Reviewer #3 (Remarks to the Author)

This is a well executed study on the physiological changes of the retinal pigmented epithelial monolayer in response to 2% apoptosis in vitro. The quality of the experimental design, the chosen assays and the execution is all excellent. The framing of the experiments is where this reviewer had much issue. There are flaws in the data interpretation and conclusions which prohibit publication or require revision. They make bodacious claims as to what their results signify. To so without qualifications of which there are many. They also use duplicitous naming of their model attempting to more closely align what they are studying (epithelial monolayer response to apoptosis) to what they claim to be studying (epithelial monolayer aging). Moreover there is no qualifications with regards to the differences between in vivo and in vitro conditions nor the known differences between iPSC-RPE compared to native RPE.

We thank the reviewer for appreciating our technical execution and experimental design and for pointing out how we could refine the interpretation of the data without overstating their significance. We have now significantly changed the manuscript as specified below.

Importantly, we replaced the abbreviation with a more appropriate one that focused on the studied phenomenon. AgeD-RPE is now density-reduced iRPE or dr-iRPE.

Additionally, we now explain how iPSC-derived RPE are an accepted model for *in vivo* RPE in lines 58-59 and discussed the limitation of our *in vitro* system for comparison with the *in vivo* native tissue in lines 415-418.

The experiments detailed in this manuscript does not support the title. "Tissue alone" is too vague for any understanding "alone" claim has not been supported by the evidence of this MS. The in vitro model is lacking many of the conditions in a live aging organism and therefore the conclusions can only be made in that model. Therefore, the title should qualify the statement. "within an in vitro model..."

Thanks to the reviewer, we believe that we now have a much more precise and impactful title. The new title is: "Cell loss disrupts mechanical homeostasis to drive retinal pigment epithelium ageing-like phenotype *in vitro*"

The abstract is too general and does not discuss what is done, what results were found and what conclusions from those results can be made.

As also pointed out from reviewer #2, we now have rewritten the abstract.

The in vitro aging may exhibit similar morphological changes for very different reasons. Lack of vascular support, lack of the ability to maintain ionic balance of the media apically and basolaterally, no differential in nutrient access apically to basolaterally. These over time can also lead to a decrease in physiological function and have nothing to do with the native environment.

Hydrogels do not have a composition resembling native RPE environment. Water content, diffusion gradient,

We agree with the reviewer that our model is highly reductionistic; therefore, conclusions cannot be drawn directly for a complex *in vivo* system. This was not sufficiently clear in the previous version of the manuscript. We have now rewritten many parts of the manuscript, clarifying that the conclusions stem from cell-density reduction in the model and not ageing. This is, for example, in lines: 409-410, 371-374, 471-473 or 494-495.

Furthermore, the difference between our model and *in vivo*, including the hydrogel substrate, is clarified in the discussion in lines 415-418 and 447-450.

The induced apoptosis is too synthetic to model native RPE aging. Therefore what is being modeled is the cellular response to apoptosis.

We agree with the reviewer and have clarified our conclusions accordingly. The revised manuscript explicitly states that our *in vitro* model does not directly replicate ageing.

However, as the model is inspired by the cell loss seen in RPE ageing *in vivo*, and because we can reproduce key features of aged RPE, we believe it is reasonable to speculate that the observed adaptation to apoptosis may highlight a previously unrecognised aspect of ageing.

There is no age dependent condition and therefore, the way the experiments are described is a misnomer.

Replace AgeD RPE with apoptosis responsive ARD-RPE and then the manuscript is on a more accurate foundation from which to draw conclusions. To mix apoptosis with age is duplicitous.

As mentioned earlier, the condition is now called density-reduced iRPE or dr-RPE, therefore avoiding any bias of interpretation.

The lack of proliferation in response to apoptosis is an interesting response that warrants further study. What is missing is a spectrum of apoptosis responses from 0 to 50% at regular intervals. Is there an apoptosis % that proliferation can be triggered? If so, what are the mechanisms governing this? Is there another more natural apoptosis approach that can support these findings or must it be the manipulation of caspase to lead to these changes within this timeframe?

These are all relevant and interesting questions, which, we think, exceed the scope of the current study and may remove the focus from our main message of a structure/mechanics-driven functional decay. Furthermore, it is technically challenging with the current method to finely modulate the apoptosis rate, and we made sure to include a control for the presence of the recombinant caspase (sup. fig. 2 and 3). Other means of cell death (e.g., staurosporin) have been initially explored with mixed success due to off-target effects.

Regarding the question of the proliferative response to varying levels of cell death, we now highlight this as a relevant aspect for future investigation in the context of important retinal pathologies (lines 404-408).

The results are informative so long as the authors are more honest with their claims.

There is sufficient detail in the methods section to reproduce the experiments.

Commend them on the proper protocol for assaying phagocytosis.

In general experimental execution and data collection quality is high.

We thank the reviewer for the positive assessment.

Point-by-point response

Reviewer #2 (Remarks to the Author)

In this revised manuscript, the authors have addressed most of the points I raised in my original review. I appreciate the authors' thorough responses.

One minor comment relates to my original point #4: Why should readers care about 'the presence of loosely interdigitated basal membranes'? The authors responded that 'We thought mentioning the loosely interdigitated basal membrane may provide a likely explanation for any questions readers may have regarding the data.' However, without context, readers (including myself) remain confused about what this observation means. To clarify this point, I suggest providing at least one specific example of a question that readers might have about your data that would be answered by this basal membrane observation and explain how this structural feature addresses that question in the text. This will help readers understand the relevance of this finding to your overall results. Since this observation is featured in both the text and figures, readers need a clear explanation of its relevance.

Once this issue is addressed, I support this manuscript for publication.

We thank the reviewer for appreciating our efforts in providing an exhaustive revised manuscript. Regarding the comment about the remaining confusion on the interdigitated basal membrane, we apologise if this remained unclear. We have now explicitly added a sentence explaining the significance of this observation for the overall interpretation of the data. This has been included in lines 274–277.

Reviewer #3 (Remarks to the Author):

The authors satisfied all of the major issues in my previous review. I support acceptance.

We thank the reviewer for the constructive comments during the previous review, which helped us clarify and strengthen the manuscript's message.